# Safe Policy Learning for Continuous Control

## Abstract

We study continuous action reinforcement learning problems in which it is crucial that the agent interacts with the environment only through *safe* policies, i.e., policies that keep the agent in desirable situations, both during training and at convergence. We formulate these problems as *constrained* Markov decision processes (CMDPs) and present safe policy optimization algorithms that are based on a *Lyapunov* approach to solve them. Our algorithms can use any standard policy gradient (PG) method, such as deep deterministic policy gradient (DDPG) or proximal policy optimization (PPO), to train a neural network policy, while guaranteeing near-constraint satisfaction for every policy update by projecting either the policy parameter or the selected action onto the set of feasible solutions induced by the state-dependent linearized Lyapunov constraints. Compared to the existing constrained PG algorithms, ours are more data efficient as they are able to utilize both on-policy and off-policy data. Moreover, our action-projection algorithm often leads to less conservative policy updates and allows for natural integration into an end-to-end PG training pipeline. We evaluate our algorithms and compare them with the state-of-the-art baselines on several simulated (MuJoCo) tasks, as well as a real-world robot obstacle-avoidance problem, demonstrating their effectiveness in terms of balancing performance and constraint satisfaction.

## 1 Introduction

The field of reinforcement learning (RL) has witnessed tremendous success in many high-dimensional control problems, including video games (Mnih et al., 2015), board games (Silver et al., 2016), robot locomotion (Lillicrap et al., 2016), manipulation (Levine et al., 2016; Kalashnikov et al., 2018), navigation (Faust et al., 2018), and obstacle avoidance (Chiang et al., 2019). In RL, the ultimate goal is to optimize the expected sum of rewards/costs, and the agent is free to explore any behavior as long as it leads to performance improvement. Although this freedom might be acceptable in many problems, including those involving simulators, and could expedite learning a good policy, it might be harmful in many other problems and could cause damage to the agent (robot) or to the environment (objects or people nearby). In such domains, it is absolutely crucial that while the agent optimizes long-term performance, it only executes safe policies both during training and at convergence.

A natural way to incorporate safety is via constraints. A standard model for RL with constraints is constrained Markov decision process (CMDP) (Altman, 1999), where in addition to its standard objective, the agent must satisfy constraints on expectations of auxiliary costs. Although optimal policies for finite CMDPs with known models can be obtained by linear programming (Altman, 1999), there are not many results for solving CMDPs when the model is unknown or the state and/or action spaces are large or infinite. A common approach to solve CMDPs is to use the Lagrangian method (Altman, 1998; Geibel & Wysotzki, 2005), which augments the original objective function with a penalty on constraint violation and computes the saddle-point of the constrained policy optimization via primal-dual methods (Chow et al., 2017). Although safety is ensured when the policy converges asymptotically, a major drawback of this approach is that it makes no guarantee with regards to the safety of the policies generated during training.

A few algorithms have been recently proposed to solve CMDPs at scale while remaining safe during training. One such algorithm is *constrained policy optimization* (CPO) (Achiam et al., 2017). CPO extends the trust-region policy optimization (TRPO) algorithm (Schulman et al., 2015a) to handle the constraints in a principled way and has shown promising empirical results in terms scalability, performance, and constraint satisfaction, both during training and at convergence. Another class of these algorithms is by Chow et al. (Chow et al., 2018). These algorithms use the notion of Lyapunov functions that have a long history in control theory to analyze the stability of dynamical systems (Khalil, 1996). Lyapunov functions have been used in RL to guarantee closed-loop stability (Perkins & Barto, 2002; Faust et al., 2014). They also have been used to guarantee that a model-based RL agent can be brought back to a "region of attraction" during exploration (Berkenkamp et al., 2017). Chow et al. (Chow et al., 2018) use the theoretical properties of the Lyapunov functions and propose safe approximate policy and value iteration algorithms. They prove theories for their algorithms when

the CMDP is finite with known dynamics, and empirically evaluate them in more general settings. However, their algorithms are value-function-based, and thus are restricted to discrete-action domains.

In this paper, we build on the problem formulation and theoretical findings of the Lyapunov-based approach to solve CMDPs, and extend it to tackle continuous action problems that play an important role in control theory and robotics. We propose Lyapunov-based safe RL algorithms that can handle problems with large or infinite action spaces, and return safe policies both during training and at convergence. To do so, there are two major difficulties that need to be addressed: **1)** the policy update becomes an optimization problem over the large or continuous action space (similar to standard MDPs with large actions), and **2)** the policy update is a constrained optimization problem in which the (Lyapunov) constraints involve integration over the action space, and thus, it is often impossible to have them in closed-form. Since the number of Lyapunov constraints is equal to the number of states, the situation is even more challenging when the problem has a large state space. To address the first difficulty, we switch from value-function-based to policy gradient (PG) algorithms. To address the second difficulty, we propose two approaches to solve our constrained policy optimization problem (a problem with infinite constraints, each involving an integral over the continuous action space) that can work with any standard on-policy (e.g., proximal policy optimization (PPO) (Schulman et al., 2017)) and off-policy (e.g., deep deterministic policy gradient (DDPG) (Lillicrap et al., 2016)) PG algorithm. Our first approach, which we call *policy parameter projection* or *θ-projection*, is a constrained optimization method that combines PG with a projection of the policy parameters onto the set of feasible solutions induced by the Lyapunov constraints. Our second approach, which we call *action projection* or *a-projection*, uses the concept of a *safety layer* introduced by (Dalal et al., 2018) to handle simple single-step constraints, extends this concept to general trajectory-based constraints, solves the constrained policy optimization problem in closed-form using Lyapunov functions, and integrates this closed-form into the policy network via safety-layer augmentation. Since both approaches guarantee safety at every policy update, they manage to maintain safety throughout training (ignoring errors resulting from function approximation), ensuring that all intermediate policies are safe to be deployed. To prevent constraint violations due to function approximation errors, similar to CPO, we offer a safeguard policy update rule that decreases constraint cost and ensures near-constraint satisfaction.

Our proposed algorithms have two main advantages over CPO. First, since CPO is closely connected to TRPO, it can only be trivially combined with PG algorithms that are regularized with relative entropy, such as PPO. This restricts CPO to on-policy PG algorithms. On the contrary, our algorithms can work with any on-policy (e.g., PPO) and off-policy (e.g., DDPG) PG algorithm. Having an off-policy implementation is beneficial, since off-policy algorithms are potentially more data-efficient, as they can use the data from the replay buffer. Second, while CPO is not a *back-propagatable* algorithm, due to the backtracking line-search procedure and the conjugate gradient iterations for computing natural gradient in TRPO, our algorithms can be trained *end-to-end*, which is crucial for scalable and efficient implementation (Hafner et al., 2017). In fact, we show in Section 3.1 that CPO (minus the line search) can be viewed as a special case of the on-policy version (PPO version) of our θ-projection algorithm, corresponding to a specific approximation of the constraints.

We evaluate our algorithms and compare them with CPO and the Lagrangian method on several continuous control (MuJoCo) tasks and a real-world robot navigation problem, in which the robot must satisfy certain constraints, while minimizing its expected cumulative cost. Results show that our algorithms outperform the baselines in terms of balancing the performance and constraint satisfaction (during training), and generalize better to new and more complex environments.

## 2 PRELIMINARIES

We consider the RL problem in which the agent's interaction with the environment is modeled as a Markov decision process (MDP). A MDP is a tuple $(\mathcal{X}, \mathcal{A}, \gamma, c, P, x_0)$, where $\mathcal{X}$ and $\mathcal{A}$ are the state and action spaces; $\gamma \in [0, 1)$ is a discounting factor; $c(x, a) \in [0, C_{\max}]$ is the immediate cost function; $P(\cdot|x, a)$ is the transition probability distribution; and $x_0 \in \mathcal{X}$ is the initial state. Although we consider deterministic initial state and cost function, our results can be easily generalized to random initial states and costs. We model the RL problems in which there are constraints on the cumulative cost using CMDPs. The CMDP model extends MDP by introducing additional costs and the associated constraints, and is defined by $(\mathcal{X}, \mathcal{A}, \gamma, c, P, x_0, d, d_0)$, where the first six components are the same as in the unconstrained MDP; $d(x) \in [0, D_{\max}]$ is the (state-dependent) immediate constraint cost; and $d_0 \in \mathbb{R}_{\geq 0}$ is an upper-bound on the expected cumulative constraint cost.

To formalize the optimization problem associated with CMDPs, let $\Delta$ be the set of Markovian stationary policies, i.e., $\Delta = \{\pi : \mathcal{X} \times \mathcal{A} \to [0, 1], \sum_a \pi(a|x) = 1\}$. At each state $x \in \mathcal{X}$, we define the generic Bellman operator w.r.t. a policy $\pi \in \Delta$ and a cost function $h$ as $T_{\pi,h}[V](x) = \sum_{a \in \mathcal{A}} \pi(a|x) [h(x, a) + \gamma \sum_{x' \in \mathcal{X}} P(x'|x, a)V(x')]$. Given a policy $\pi \in \Delta$, we define the expected

cumulative cost and the safety constraint function (expected cumulative constraint cost) as $\mathcal{C}_\pi(x_0) := \mathbb{E}[\sum_{t=0}^\infty \gamma^t c(x_t, a_t) \mid \pi, x_0]$ and $\mathcal{D}_\pi(x_0) := \mathbb{E}[\sum_{t=0}^\infty \gamma^t d(x_t) \mid \pi, x_0]$, respectively. The *safety constraint* is then defined as $\mathcal{D}_\pi(x_0) \le d_0$. The goal in CMDPs is to solve the constrained optimization problem

$$\pi^* \in \arg\min_{\pi \in \Delta} \{ \mathcal{C}_\pi(x_0) : \mathcal{D}_\pi(x_0) \le d_0 \}. \tag{1}$$

It has been shown that if the feasibility set is non-empty, then there exists an optimal policy in the class of stationary Markovian policies $\Delta$ (Altman, 1999, Theorem 3.1).

## 2.1 POLICY GRADIENT ALGORITHMS

Policy gradient (PG) algorithms optimize a policy by computing a sample estimate of the gradient of the expected cumulative cost induced by the policy, and then updating the policy parameter in the gradient direction. In general, stochastic policies that give a probability distribution over actions are parameterized by a $\kappa$-dimensional vector $\theta$, so the space of policies can be written as $\{\pi_\theta, \ \theta \in \Theta \subset \mathbb{R}^\kappa\}$. Since in this setting a policy $\pi$ is uniquely defined by its parameter $\theta$, policy-dependent functions can be written as a function of $\theta$ or $\pi$ interchangeably.

DDPG (Lillicrap et al., 2016) and PPO (Schulman et al., 2017) are two PG algorithms that have recently gained popularity in solving continuous control problems. DDPG is an off-policy Q-learning style algorithm that jointly trains a deterministic policy $\pi_\theta(x)$ and a Q-value approximator $Q(x, a; \phi)$. The Q-value approximator is trained to fit the true Q-value function and the deterministic policy is trained to optimize $Q(x, \pi_\theta(x); \phi)$ via chain-rule. The PPO algorithm we use in this paper is a penalty form of TRPO (Schulman et al., 2015a) with an adaptive rule to tune the $D_{KL}$ penalty weight $\beta_k$. PPO trains a policy $\pi_\theta(x)$ by optimizing a loss function that consists of the standard policy gradient objective and a penalty on the KL-divergence between the current $\theta$ and previous $\theta'$ policies, i.e., $\overline{D}_{\text{KL}}(\theta, \theta') = \mathbb{E}[\sum_t \gamma^t D_{\text{KL}}(\pi_{\theta'}(\cdot|x_t)||\pi_\theta(\cdot|x_t))|\pi_{\theta'}, x_0]$.

## 2.2 LAGRANGIAN METHOD

Lagrangian method is a straightforward way to address the constraint $\mathcal{D}_{\pi_\theta}(x_0) \le d_0$ in CMDPs. Lagrangian method adds the constraint costs $d(x)$ to the task costs $c(x, a)$ and transform the constrained optimization problem to a penalty form, i.e., $\min_{\theta \in \Theta} \max_{\lambda \ge 0} \mathbb{E}[\sum_{t=0}^\infty c(x_t, a_t) + \lambda d(x_t)|\pi_\theta, x_0] - \lambda d_0$. The method then jointly optimizes $\theta$ and $\lambda$ to find a saddle-point of the penalized objective. The optimization of $\theta$ may be performed by any PG algorithm on the augmented cost $c(x, a) + \lambda d(x)$, while $\lambda$ is optimized by stochastic gradient descent. As described in Sec. 1, although the Lagrangian approach is easy to implement (see Appendix A for the details), in practice, it often violates the constraints during training. While at each step during training, the objective encourages finding a safe solution, the current value of $\lambda$ may lead to an unsafe policy. This is why the Lagrangian method may not be suitable for solving problems in which safety is crucial during training.

## 2.3 LYAPUNOV FUNCTIONS

Since in this paper, we extend the Lyapunov-based approach to CMDPs of (Chow et al., 2018) to PG algorithms, we end this section by introducing some terms and notations from (Chow et al., 2018) that are important in developing our safe PG algorithms. We refer readers to Appendix B for details.

We define a set of Lyapunov functions w.r.t. initial state $x_0 \in \mathcal{X}$ and constraint threshold $d_0$ as $\mathcal{L}_{\pi_B}(x_0, d_0) = \{L : \mathcal{X} \to \mathbb{R}_{\ge 0} \mid T_{\pi_B, d}[L](x) \le L(x), \ \forall x \in \mathcal{X}, \ L(x_0) \le d_0\}$, where $\pi_B$ is a feasible policy of (1), i.e., $\mathcal{D}_{\pi_B}(x_0) \le d_0$. We refer to the constraints in this feasibility set as the *Lyapunov constraints*. For an arbitrary Lyapunov function $L \in \mathcal{L}_{\pi_B}(x_0, d_0)$, we denote by $\mathcal{F}_L = \{\pi \in \Delta : T_{\pi, d}[L](x) \le L(x), \ \forall x \in \mathcal{X}\}$, the set of $L$-induced Markov stationary policies. The contraction property of $T_{\pi, d}$, together with $L(x_0) \le d_0$, imply that any $L$-induced policy in $\mathcal{F}_L$ is a feasible policy of (1). However, $\mathcal{F}_L(x)$ does not always contain an optimal solution of (1), and thus, it is necessary to design a Lyapunov function that provides this guarantee. In other words, the main goal of the Lyapunov approach is to construct a Lyapunov function $L \in \mathcal{L}_{\pi_B}(x_0, d_0)$, such that $\mathcal{F}_L$ contains an optimal policy $\pi^*$, i.e., $L(x) \ge T_{\pi^*, d}[L](x)$. Chow et al. (2018) show in their Theorem 1 that without loss of optimality, the Lyapunov function that satisfies the above criterion can be expressed as $L_{\pi_B, \epsilon}(x) := \mathbb{E}\big[\sum_{t=0}^\infty \gamma^t(d(x_t) + \epsilon(x_t)) \mid \pi_B, x\big]$, in which $\epsilon(x) \ge 0$ is a specific immediate *auxiliary constraint cost* that keeps track of the maximum *constraint budget* available for policy improvement (from $\pi_B$ to $\pi^*$). They propose ways to construct such $\epsilon$, as well as an auxiliary constraint cost surrogate $\widetilde{\epsilon}$, which is a tight upper-bound on $\epsilon$ and can be computed more efficiently. They use this construction to propose the safe (approximate) policy and value iteration algorithms, whose objective is to solve the following LP (Chow et al., 2018, Eq. 6) during policy improvement:

$$\pi_+(\cdot|x) = \arg\min_{\pi \in \Delta} \int_{a \in \mathcal{A}} Q_{V_{\pi_B}}(x, a)\pi(a|x), \quad \text{s.t.} \int_{a \in \mathcal{A}} Q_{L_{\pi_B}}(x, a) \left(\pi(a|x) - \pi_B(a|x)\right) \le \widetilde{\epsilon}(x), \tag{2}$$

where $V_{\pi_B}(x) = T_{\pi_B,c}[V_{\pi_B}](x)$ and $Q_{V_{\pi_B}}(x,a) = c(x,a) + \gamma \sum_{x'} P(x'|x,a)V_{\pi_B}(x')$ are the value and state-action value functions (w.r.t. the cost function $c$), and $Q_{L_{\pi_B}}(x,a) = d(x) + \widetilde{\epsilon}(x) + \gamma \sum_{x'} P(x'|x,a)L_{\pi_B,\widetilde{\epsilon}}(x')$ is the Lyapunov function. In any iterative policy optimization method, such as those studied in this paper, the feasible policy $\pi_B$ at each iteration can be set to the policy computed at the previous iteration (which is feasible).

In LP (2), there are as many constraints as the number of states and each constraint involves an integral over the entire action space. When the state space is large, even if the integral in the constraint has a closed-form (e.g., for finite actions), solving (2) becomes numerically intractable. Chow et al. (Chow et al., 2018) assumed that the number of actions is finite and focused on value-function-based RL algorithms, and addressed the large state issue by *policy distillation*. Since in this paper, we are interested in problems with large action spaces, solving (2) will be even more challenging. To address this issue, in the next section, we first switch from value-function-based algorithms to PG algorithms, then propose an optimization problem with Lyapunov constraints, analogous to (2), that is suitable for PG, and finally present two methods to solve our proposed optimization problem efficiently.

## 3 SAFE LYAPUNOV-BASED POLICY GRADIENT

We now present our approach to solve CMDPs in a way that guarantees safety both at convergence and during training. Similar to (Chow et al., 2018), our Lyapunov-based safe PG algorithms solve a constrained optimization problem analogous to (2). In particular, our algorithms consist of two components, a baseline PG algorithm, such as DDPG or PPO, and an effective method to solve the general Lyapunov-based policy optimization problem, the analogous to (2), i.e,

$$\theta_+ = \underset{\theta \in \Theta}{\arg\min} \; \mathcal{C}_{\pi_\theta}(x_0), \quad \text{s.t.} \int_{a \in \mathcal{A}} \left(\pi_\theta(a|x) - \pi_B(a|x)\right) Q_{L_{\pi_B}}(x,a) \, da \leq \widetilde{\epsilon}(x), \;\; \forall x \in \mathcal{X}. \quad (3)$$

In the next two sections, we present two approaches to solve (3) efficiently. We call these approaches **1)** $\theta$-projection, a constrained optimization method that combines PG with projecting the policy parameter $\theta$ onto the set of feasible solutions induced by the Lyapunov constraints, and **2)** $a$-projection, in which we embed the Lyapunov constraints into the policy network via a safety layer.

### 3.1 THE $\theta$-PROJECTION APPROACH

The $\theta$-projection approach is based on the *minorization-maximization* technique in conservative PG (Kakade & Langford, 2002) and Taylor series expansion, and can be applied to both on-policy and off-policy algorithms. Following Theorem 4.1 in (Kakade & Langford, 2002), we first have the following bound for the cumulative cost: $-\beta \overline{D}_{\mathrm{KL}}(\theta, \theta_B) \leq \mathcal{C}_{\pi_\theta}(x_0) - \mathcal{C}_{\pi_{\theta_B}}(x_0) - \mathbb{E}_{x \sim \mu_{\theta_B,x_0}, a \sim \pi_\theta}[Q_{V_{\theta_B}}(x,a) - V_{\theta_B}(x)] \leq \beta \overline{D}_{\mathrm{KL}}(\theta, \theta_B)$, where $\mu_{\theta_B,x_0}$ is the $\gamma$-visiting distribution of $\pi_{\theta_B}$ starting at the initial state $x_0$, and $\beta$ is the weight for the entropy-based regularization.[1] Using this result, we denote by $\mathcal{C}'_{\pi_\theta}(x_0; \pi_{\theta_B}) = \mathcal{C}_{\pi_{\theta_B}}(x_0) + \beta \overline{D}_{\mathrm{KL}}(\theta, \theta_B) + \mathbb{E}_{x \sim \mu_{\theta_B,x_0}, a \sim \pi_\theta}[Q_{V_{\theta_B}}(x,a) - V_{\theta_B}(x)]$ the surrogate cumulative cost. It has been shown in Eq. 10 of (Schulman et al., 2015a) that replacing the objective function $\mathcal{C}_{\pi_\theta}(x_0)$ with its surrogate $\mathcal{C}'_{\pi_\theta}(x_0; \pi_{\theta_B})$ in solving (3) will still lead to policy improvement. In order to effectively compute the improved policy parameter $\theta_+$, one further approximates the function $\mathcal{C}'_{\pi_\theta}(x_0; \pi_{\theta_B})$ with its Taylor series expansion around $\theta_B$. In particular, the term $\mathbb{E}_{x \sim \mu_{\theta_B,x_0}, a \sim \pi_\theta}[Q_{V_{\theta_B}}(x,a) - V_{\theta_B}(x)]$ is approximated up to its first order, and the term $\overline{D}_{\mathrm{KL}}(\theta, \theta_B)$ is approximated up to its second order. These altogether allow us to replace the objective function in (3) with $\langle(\theta - \theta_B), \nabla_\theta \mathbb{E}_{x \sim \mu_{\theta_B,x_0}, a \sim \pi_\theta}[Q_{V_{\theta_B}}(x,a)]\rangle + \frac{\beta}{2}\langle(\theta - \theta_B), \nabla_\theta^2 \overline{D}_{\mathrm{KL}}(\theta, \theta_B) \mid_{\theta=\theta_B} (\theta - \theta_B)\rangle$.

Similarly, regarding the constraints in (3), we can use the Taylor series expansion (around $\theta_B$) to approximate the LHS of the Lyapunov constraints as $\int_{a \in \mathcal{A}}(\pi_\theta(a|x) - \pi_B(a|x)) Q_L(x,a) \, da \approx \langle(\theta - \theta_B), \nabla_\theta \mathbb{E}_{a \sim \pi_\theta}[Q_{L_{\theta_B}}(x,a)] \mid_{\theta=\theta_B}\rangle$. Using the above approximations, at each iteration, our safe PG algorithm updates the policy by solving the following constrained optimization problem with *semi-infinite dimensional* Lyapunov constraints:

$$\theta_+ \in \underset{\theta \in \Theta}{\arg\min} \; \langle(\theta - \theta_B), \nabla_\theta \mathbb{E}_{x \sim \mu_{\theta_B,x_0}, a \sim \pi_\theta}[Q_{V_{\theta_B}}(x,a)]\rangle + \frac{\beta}{2}\langle(\theta - \theta_B), \nabla_\theta^2 \overline{D}_{\mathrm{KL}}(\theta, \theta_B) \mid_{\theta=\theta_B} (\theta - \theta_B)\rangle,$$

$$\text{s.t.} \; \langle(\theta - \theta_B), \nabla_\theta \mathbb{E}_{a \sim \pi_\theta}[Q_{L_{\theta_B}}(x,a)] \mid_{\theta=\theta_B} \rangle \leq \widetilde{\epsilon}(x), \; \forall x \in \mathcal{X}. \quad (4)$$

It can be seen that if the errors resulted from the neural network parameterizations of $Q_{V_{\theta_B}}$ and $Q_{L_{\theta_B}}$, and the Taylor series expansions are small, then an algorithm that updates the policy parameter by solving (4) can ensure safety during training. However, the presence of infinite-dimensional Lyapunov constraints makes solving (4) intractable. A solution to this is to write the Lyapunov constraints in (4) (without loss of optimality) as $\max_{x \in \mathcal{X}}\langle(\theta - \theta_B), \nabla_\theta \mathbb{E}_{a \sim \pi_\theta}[Q_{L_{\theta_B}}(x,a)] \mid_{\theta=\theta_B}\rangle - \widetilde{\epsilon}(x) \leq 0$. Since

---

[1]Theorem 1 in (Schulman et al., 2015a) provides a recipe for computing $\beta$ such that the minorization-maximization inequality holds. But in practice, $\beta$ is treated as a tunable parameter for entropy regularization.

the above $\max$-operator is non-differentiable, this may still lead to numerical instability in gradient descent algorithms. Similar to the surrogate constraint in TRPO (to transform the $\max D_{\mathrm{KL}}$ constraint to an average $\overline{D}_{\mathrm{KL}}$ constraint, see Eq. 12 in (Schulman et al., 2015a)), a more numerically stable way is to *approximate* the Lyapunov constraint using the average constraint surrogate

$$\big\langle (\theta - \theta_B), \frac{1}{M} \sum_{i=1}^{M} \nabla_\theta \mathbb{E}_{a \sim \pi_\theta}[Q_{L_{\theta_B}}(x_i, a)] \mid_{\theta = \theta_B} \big\rangle \leq \frac{1}{M} \sum_{i=1}^{M} \widetilde{\epsilon}(x_i), \tag{5}$$

where $M$ is the number of on-policy sample trajectories of $\pi_{\theta_B}$. In order to effectively compute the gradient of the Lyapunov value function, consider the special case when the auxiliary constraint surrogate is chosen as $\widetilde{\epsilon} = (1 - \gamma)(d_0 - \mathcal{D}_{\pi_{\theta_B}}(x_0))$ (see Appendix B for justification). Using the fact that $\widetilde{\epsilon}$ is $\theta$-independent, the gradient term in (5) can be written as $\int_a \pi_\theta(a|x) \nabla_\theta \log \pi_\theta(a|x) Q_{W_{\theta_B}}(x_i, a) da$, where $W_{\theta_B}(x) = T_{\pi_B, d}[W_{\theta_B}](x)$ and $Q_{W_{\theta_B}}(x, a) = d(x) + \gamma \sum_{x'} P(x'|x, a) W_{\theta_B}(x')$ are the constraint value functions, respectively. Since the integral is equal to $E_{a \sim \pi_\theta}[Q_{W_{\theta_B}}(x_i, a)]$, the average constraint surrogate (5) can be approximated (approximation is because of the choice of $\widetilde{\epsilon}$) by the inequality $\mathcal{D}_{\pi_{\theta_B}}(x_0) + \frac{1}{1-\gamma} \langle (\theta - \theta_B), \frac{1}{M} \sum_{i=1}^{M} \nabla_\theta \mathbb{E}_{a \sim \pi_\theta}[Q_{W_{\theta_B}}(x_i, a)] |_{\theta = \theta_B} \rangle \leq d_0$, which is equivalent to the constraint used in CPO (see Section 6.1 in (Achiam et al., 2017)). This shows that CPO (minus the line search) belongs to the class of our Lyapunov-based PG algorithms with $\theta$-projection. We refer to the DDPG and PPO versions of our $\theta$-projection safe PG algorithms as SDDPG and SPPO. Derivation details and the pseudo-code (Algorithm 4) of these algorithms are given in Appendix C.

## 3.2 THE $a$-PROJECTION APPROACH

The main characteristic of the Lyapunov approach is to break down a trajectory-based constraint into a sequence of single-step *state dependent* constraints. However, when the state space is infinite, the feasibility set is characterized by infinite dimensional constraints, and thus, it is counter-intuitive to directly enforce these Lyapunov constraints (as opposed to the original trajectory-based constraint) into the policy update optimization. To address this, we leverage the idea of a *safety layer* from (Dalal et al., 2018), that was applied to simple single-step constraints, and propose a novel approach to embed the set of Lyapunov constraints into the policy network. This way, we reformulate the CMDP problem (1) as an unconstrained optimization problem and optimize its policy parameter $\theta$ (of the augmented network) using any standard unconstrained PG algorithm. At every given state, the unconstrained action is first computed and then passed through the safety layer, where a feasible action mapping is constructed by projecting unconstrained actions onto the feasibility set w.r.t. Lyapunov constraints. This *constraint projection approach* can guarantee safety during training.

We now describe how the action mapping (to the set of Lyapunov constraints) works[2]. Recall from the policy improvement problem in (3) that the Lyapunov constraint is imposed at every state $x \in \mathcal{X}$. Given a baseline feasible policy $\pi_B = \pi_{\theta_B}$, for any arbitrary policy parameter $\theta \in \Theta$, we denote by $\Xi(\pi_B, \theta) = \{\theta' \in \Theta : Q_{L_{\pi_B}}(x, \pi_{\theta'}(x)) - Q_{L_{\pi_B}}(x, \pi_B(x)) \leq \widetilde{\epsilon}(x), \forall x \in \mathcal{X}\}$, the *projection* of $\theta$ onto the feasibility set induced by the Lyapunov constraints. One way to construct a feasible policy $\pi_{\Xi(\pi_B, \theta)}$ from a parameter $\theta$ is to solve the following $\ell_2$-projection problem:

$$\pi_{\Xi(\pi_B, \theta)}(x) \in \arg\min_{a \in \mathcal{A}} \frac{1}{2} \|a - \pi_\theta(x)\|^2, \quad \text{s.t.} \quad Q_{L_{\pi_B}}(x, a) - Q_{L_{\pi_B}}(x, \pi_B(x)) \leq \widetilde{\epsilon}(x). \tag{6}$$

We refer to this operation as the *Lyapunov safety layer*. Intuitively, this projection perturbs the unconstrained action as little as possible in the Euclidean norm in order to satisfy the Lyapunov constraints. Since this projection guarantees safety, if we have access to a closed form of the projection, we may insert it into the policy parameterization and simply solve an unconstrained policy optimization problem, i.e., $\theta_+ \in \arg\min_{\theta \in \Theta} \mathcal{C}_{\pi_{\Xi(\pi_B, \theta)}}(x_0)$, using any standard PG algorithm.

To simplify the projection (6), we can approximate the LHS of the Lyapunov constraint with its first-order Taylor series (w.r.t. action $a = \pi_B(x)$). Thus, at any given state $x \in \mathcal{X}$, the safety layer solves the following projection problem:

$$\pi_{\Xi(\pi_B, \theta)}(x) \in \arg\min_{a \in \mathcal{A}} \frac{1 - \eta(x)}{2} \|a - \pi_\theta(x)\|^2 + \frac{\eta(x)}{2} \|a - \pi_B(x)\|^2, \text{ s.t. } (a - \pi_B(x))^\top g_{L_{\pi_B}}(x) \leq \widetilde{\epsilon}(x), \tag{7}$$

where $\eta(x) \in [0, 1)$ is the mixing parameter that controls the trade-off between projecting on unconstrained policy (for return maximization) and on baseline policy (for safety), and $g_{L_{\pi_B}}(x) := \nabla_a Q_{L_{\pi_B}}(x, a) |_{a = \pi_B(x)}$ is the action-gradient of the state-action Lyapunov function.

Similar to the analysis of Section 3.1, if the auxiliary cost $\widetilde{\epsilon}$ is state-independent, one can readily find $g_{L_{\pi_B}}(x)$ by computing the gradient of the constraint action-value function $\nabla_a Q_{W_{\theta_B}}(x, a) |_{a = \pi_B(x)}$.

---

[2]In our experiments, we use stochastic (Gaussian) policies with parameterized mean and fixed variance. We leave extension of the $a$-projection approach to policies in which variance is also parameterized as future work.

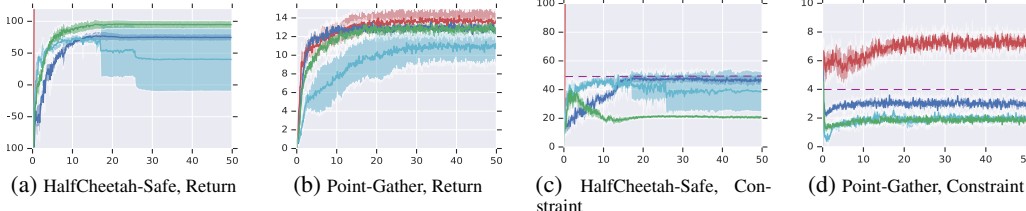

| (a) HalfCheetah-Safe, Return | (b) Point-Gather, Return | (c) HalfCheetah-Safe, Constraint | (d) Point-Gather, Constraint |

Figure 1: DDPG (red), DDPG-Lagrangian (cyan), SDDPG (blue), SDDPG $a$-projection (green) on HalfCheetah-Safe and Point-Gather. SDDPG and SDDPG $a$-projection perform stable and safe learning, although the dynamics and cost functions are unknown, control actions are continuous, and deep function approximations are used. Unit of x-axis is in thousands of episodes. Shaded areas represent the 1-SD confidence intervals (over 10 random seeds). The dashed purple line in the two right figures represents the constraint limit.

Note that the objective function in (7) is positive-definite and quadratic, and the constraint approximation is linear. Therefore, the solution of this (convex) projection problem can be effectively computed by an in-graph QP-solver, such as OPT-Net (Amos & Kolter, 2017). Combined with the above projection procedure, this further implies that the CMDP problem can be effectively solved using an *end-to-end* PG training pipeline (such as DDPG or PPO). When the CMDP has a single constraint (and thus a single Lyapunov constraint), the policy $\pi_{\Xi(\pi_B,\theta)}(x)$ has the following analytical solution.

**Proposition 1.** *At any given state* $x \in \mathcal{X}$, *the solution to the optimization problem* (7) *has the form* $\pi_{\Xi(\pi_B,\theta)}(x) = (1 - \eta(x))\pi_\theta(x) + \eta(x)\pi_B(x) - \lambda^*(x) \cdot g_{L_{\pi_B}}(x)$, *where* $\lambda^*(x) = \left(\left((1 - \eta(x)) \cdot g_{L_{\pi_B}}(x)^\top(\pi_\theta(x) - \pi_B(x)) - \widetilde{\epsilon}(x)\right)/g_{L_{\pi_B}}(x)^\top g_{L_{\pi_B}}(x)\right)_+$.

The closed-form solution is essentially a linear projection of the unconstrained action $\pi_\theta(x)$ onto the Lyapunov-safe hyper-plane with slope $g_{L_{\pi_B}}(x)$ and intercept $\widetilde{\epsilon}(x) = (1 - \gamma)(d_0 - \mathcal{D}_{\pi_B}(x_0))$. It is possible to extend this closed-form solution to handle multiple constraints, if there is at most one constraint active at a time (see Proposition 1 in (Dalal et al., 2018)). We refer to the DDPG and PPO versions of our $a$-projection safe Lyapunov-based PG algorithms as SDDPG $a$-projection and SPPO $a$-projection. Derivation and pseudo-code (Algorithm 5) of these algorithms are in Appendix C.

## 4 EXPERIMENTS ON MUJOCO BENCHMARKS

We empirically evaluate[3] our Lyapunov-based safe PG algorithms to assess their: (i) performance in terms of cost and safety during training, and (ii) robustness w.r.t. constraint violation. We use three simulated robot locomotion continuous control tasks in the MuJoCo simulator (Todorov et al., 2012). The notion of safety in these tasks is motivated by physical constraints: (i) HalfCheetah-Safe: this is a modification of the MuJoCo HalfCheetah problem in which we impose constraints on the speed of Cheetah in order to force it to run smoothly. The video shows that the policy learned by our algorithm results in slower but much smoother movement of Cheetah compared to the policies learned by PPO and Lagrangian[4]; (ii) Point-Circle: the agent is rewarded for running in a wide circle, but is constrained to stay within a safe region defined by $|x| \le x_{\lim}$; (iii) Point-Gather & Ant-Gather: the agent is rewarded for collecting target objects in a terrain map, while being constrained to avoid bombs. The last two tasks were first introduced in (Achiam et al., 2017) by adding constraints to the original MuJoCo tasks: *Point* and *Ant*. Details of these tasks are given in Appendix D.

We compare our algorithms with two state-of-the-art unconstrained algorithms, DDPG and PPO, and two constrained methods, Lagrangian with optimized Lagrange multiplier (Appendix A) and on-policy CPO. We use the CPO algorithm that is based on PPO (unlike the original CPO that is based on TRPO) and coincides with our SPPO algorithm derived in Section 4.1. SPPO preserves the essence of CPO by adding the first-order constraint and relative entropy regularization to the policy optimization problem. The main difference between CPO and SPPO is that the latter *does not* perform backtracking line-search in learning rate. We compare with SPPO instead of CPO to **1)** avoid the additional computational complexity of line-search in TRPO, while maintaining the performance of PG using PPO, **2)** have a back-propagatable version of CPO, and **3)** have a fair comparison with other back-propagatable safe PG algorithms, such as our DDPG and $a$-projection based algorithms.

**Comparison with baselines:** Figures 1a, 1b, 2a, 2b, 8a, 8b, 9a, 9b show that our Lyapunov-based PG algorithms are stable in learning and all converge to feasible policies with reasonable performance. Figures 1c, 1d, 2c, 2d, 8c, 8d, 9c, 9b show the algorithms in terms of constraint violation during

---

[3]Videos of MuJoCo experiments can be found at `https://drive.google.com/file/d/1FwbuEnKN2lLWFMKvDCydo2EQVdc14O1a/view?usp=sharing`.

[4]We also imposed constraint on the torque at the Cheetah's joints in order to force it to run more smoothly and obtained similar results as imposing constraint on its speed.

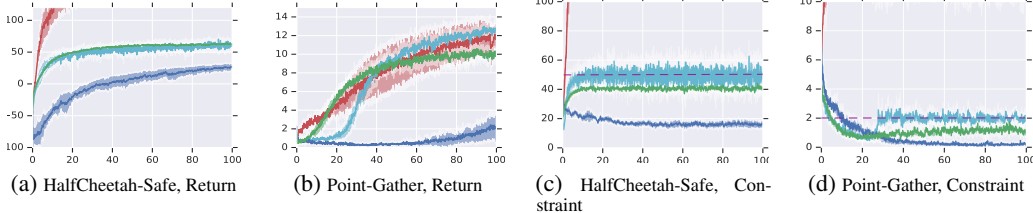

(a) HalfCheetah-Safe, Return    (b) Point-Gather, Return    (c) HalfCheetah-Safe, Constraint    (d) Point-Gather, Constraint

Figure 2: PPO (red), PPO-Lagrangian (cyan), SPPO (blue), SPPO $a$-projection (green) on HalfCheetah-Safe and Point-Gather. SPPO $a$-projection perform stable and safe learning, when the dynamics and cost functions are unknown, control actions are continuous, and deep function approximation is used.

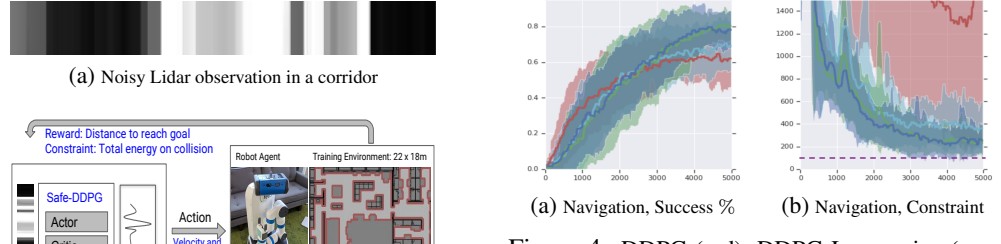

(a) Noisy Lidar observation in a corridor

(b) SDDPG for point to point task

Figure 3: Robot navigation task details.

(a) Navigation, Success %    (b) Navigation, Constraint

Figure 4: DDPG (red), DDPG-Lagrangian (cyan), SDDPG (blue), DDPG $a$-projection (green) on Robot Navigation. Ours (SDDPG, SDDPG $a$-projection) balance between reward and constraint learning. Unit of x-axis is in thousands of steps. The shaded areas represent the 1-SD confidence intervals (over 50 runs). The dashed purple line represents the constraint limit.

training. These figures indicate that our algorithms quickly stabilize the constraint cost below the threshold, while the unconstrained DDPG and PPO violate the constraints, and Lagrangian tends to jiggle around the threshold. Moreover, it is worth-noting that the Lagrangian method can be sensitive to the initialization of the Lagrange multiplier $\lambda_0$. If $\lambda_0$ is too large, it would make policy updates overly conservative, and if it is too small, then we will have more constraint violation. Without further knowledge about the environment, we treat $\lambda_0$ as a hyper-parameter and optimize it via grid-search. See Appendix D for more details and for the experimental results of *Ant-Gather* and *Point-Circle*.

$a$**-projection vs. $\theta$-projection:** The figures indicate that in many cases DDPG and PPO with $a$-projection converge faster and have lower constraint violation than their $\theta$-projection counterparts (i.e., SDDPG and SPPO). This corroborates with the hypothesis that $a$-projection is less conservative during policy updates than $\theta$-projection (which is what CPO is based on) and generates smoother gradient updates during end-to-end training.

**DDPG vs. PPO:** In most experiments (HalfCheetah, PointGather, and AntGather) the DDPG algorithms tend to have faster learning than their PPO counterparts, while the PPO algorithms perform better in terms of constraint satisfaction. The faster learning behavior is due to the improved data-efficiency when using off-policy samples in PG, however, the covariate-shift [5] in off-policy data makes tight constraint control more challenging.

## 5 SAFE POLICY GRADIENT FOR ROBOT NAVIGATION

We now evaluate safe policy optimization algorithms on a real robot task – a *map-less* navigation task (Chiang et al., 2019) – where a noisy differential drive robot with limited sensors (Fig. 3a) is required to navigate to a goal outside of its field of view in unseen environments while avoiding collision. The main goal is to learn a policy that drives the robot to goal as efficiently as possible, while limiting the impact energy of collisions, since the collision can damage the robot and environment.

Here the CMDP is non-discounting and has a fixed horizon. The agent's observations consist of the relative goal position, agent's velocity, and Lidar measurements (Fig. 3a). The actions are the linear and angular velocity at the robot's center of the mass. [6] The transition probability captures the noisy robot's dynamics, whose exact formulation is unknown to the robot. The robot must navigate

---

[5]Here covariate shift is due to the fact that training data is generated by a policy that is different from the current policy that is being optimized.

[6]The first dimension is the robot's desired linear velocity (speed at which the robot should go straight). The second dimension is the robot's angular velocity - speed at which the robot should turn. Both velocity vectors are applied on the center of the mass of the robot.

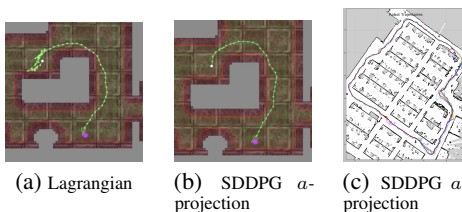

(a) Lagrangian    (b) SDDPG $a$-projection    (c) SDDPG $a$-projection

Figure 5: Navigation routes of two learned policies in the simulator (a) and (b). On-robot experiment (c).

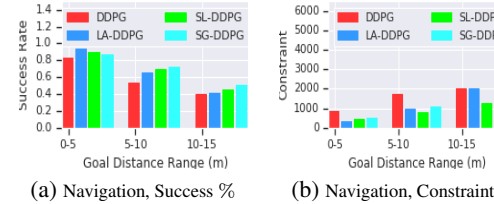

(a) Navigation, Success %      (b) Navigation, Constraint

Figure 6: Generalization over success rate (d) and constraint satisfaction (e) on a different environment. The average success rate and cumulative (trajectory-based) constraint cost are over 100 tasks (randomly sampled start and goal robot positions). A task is considered successful if the robot reaches the goal in the map-less navigation task, regardless of the constraint.

to arbitrary goal positions collision-free in a previously unseen environment, and without access to the indoor map and any work-space topology. We reward the agent for reaching the goal, which translates to an immediate cost that measures the relative distance to the goal. To measure the total impact energy of obstacle collisions, we impose an immediate constraint cost to account for the speed during collision, with a constraint threshold $d_0$ that characterizes the agent's maximum tolerable collision impact energy to any object. Different from the standard approach, where a constraint on collision speed is explicitly imposed to the learning problem at each time step, we emphasize that a CMDP constraint is required here because it allows the robot to lightly brush off the obstacle (such as walls) but prevent it from ramming into any objects. Other use cases of CMDP constraints in robot navigation include collision avoidance (Pfeiffer et al., 2018) or limiting total battery usage of the task.

**Experimental Results:** We evaluate the learning algorithms on success rate and constraint control averaged over 100 episodes with random initialization. The task is successful if the robot reaches the goal before the constraint threshold (total energy of collision) is exhausted. While all methods converge to policies with reasonable performance, Figure 4a and 4b show that the Lyapunov-based PG algorithms have higher success rates, due to their robust abilities of controlling the total constraint, as well minimizing the distance to goal. Although the unconstrained method often yields a lower distance to goal, it violates the constraint more frequently leading to a lower success rate. Lagrangian approach is less robust to initialization of parameters, and therefore it generally has lower success rate and higher variability than the Lyapunov-based methods. Unfortunately due to function approximation error and stochasticity of the problem, all the algorithms converged pre-maturely with constraints above the threshold, possibly due to the overly conservative constraint threshold ($d_0 = 100$). Inspection of trajectories shows that the Lagrangian method tends to zigzag and has more collisions, while the SDDPG chooses a safer path to reach the goal (Figures 5a and 5b).

Next, we evaluate how well the methods generalize to (i) longer trajectories, and (ii) new environments. The tasks are trained in a 22 by 18 meters environment (Fig. 7) with goals placed within 5 to 10 meters from the robot initial state. In a much larger evaluation environment (60 by 47 meters) with goals placed up to 15 meters away from the goal, the success rate of all methods degrades as the goals are further away (Fig. 6a). The safety methods ($a$-projection – SL-DDPG, and $\theta$-projection – SG-DDPG) outperform unconstrained and Lagrangian (DDPG and LA-DDPG), while retaining the lower constraints even when the task becomes more difficult (Fig. 6b).

Finally, we deployed the SL-DDPG policy onto the real Fetch robot (Wise et al., 2016) in an everyday office environment. [7] Fetch robot weights 150 kilograms, and reaches maximum speed of 7 km/h making the collision force a safety paramount. Figure 5c shows the top down view of the robot log. Robot travelled, through narrow corridors and around people walking through the office, for a total of 500 meters to complete five repetitions of 12 tasks, each averaging about 10 meters to the goal. The robot robustly avoids both static and dynamic (humans) obstacles coming into its path. We observed additional "wobbling" effects, that was not present in simulation. This is likely due to the wheel slippage at the floor that the policy was not trained for. In several occasions when the robot could not find a clear path, the policy instructed the robot to stay put instead of narrowly passing by the obstacle. This is precisely the safety behavior we want to achieve with the Lyapunov-based algorithms.

## 6 CONCLUSIONS AND FUTURE WORK

We used the notion of Lyapunov functions and developed a class of safe RL algorithms for continuous action problems. Each algorithm in this class is a combination of one of our two proposed projections:

---

[7]Videos of Fetch robot navigation can be found in the following link: `https://drive.google.com/file/d/1FwbuEnKN2lLWFMKvDCydo2EQVdc14O1a/view?usp=sharing`

$\theta$-projection and $a$-projection, with any on-policy (e.g., PPO) or off-policy (e.g., DDPG) PG algorithm. We evaluated our algorithms on four high-dimensional simulated robot locomotion MuJoCo tasks and compared them with several baselines. To demonstrate the effectiveness of our algorithms in solving real-world problems, we also applied them to an indoor robot navigation problem, to ensure that the robot's path is optimal and collision-free. Our results indicate that our algorithms **1)** achieve safe learning, **2)** have better data-efficiency, **3)** can be more naturally integrated within the standard end-to-end differentiable PG training pipeline, and **4)** are scalable to tackle real-world problems. Our work is a step forward in deploying RL to real-world problems in which safety guarantees are of paramount importance. Future work includes **1)** extending $a$-projection to stochastic policies and **2)** extensions of the Lyapunov approach to model-based RL and use it for safe exploration.

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

## A  THE LAGRANGIAN APPROACH TO SAFE RL

We first state a number of mild technical and notational assumptions that we make throughout this section.

**Assumption 1** (Differentiability). *For any state-action pair $(x, a)$, $\pi_\theta(a|x)$ is continuously differentiable in $\theta$ and $\nabla_\theta \pi_\theta(a|x)$ is a Lipschitz function in $\theta$ for every $x \in \mathcal{X}$ and $a \in \mathcal{A}$.*

**Assumption 2** (Strict Feasibility). *There exists a transient policy $\pi_\theta(\cdot|x)$ such that $\mathcal{D}_{\pi_\theta}(x_0) < d_0$ in the constrained problem.*

**Assumption 3** (Step Sizes). *The step size schedules $\{\alpha_{3,k}\}$, $\{\alpha_{2,k}\}$, and $\{\alpha_{1,k}\}$ satisfy*

$$\sum_k \alpha_{1,k} = \sum_k \alpha_{2,k} = \sum_k \alpha_{3,k} = \infty, \tag{8}$$

$$\sum_k \alpha_{1,k}^2, \quad \sum_k \alpha_{2,k}^2, \quad \sum_k \alpha_{3,k}^2 < \infty, \tag{9}$$

$$\alpha_{1,k} = o(\alpha_{2,k}), \quad \alpha_{2,k} = o(\alpha_{3,k}). \tag{10}$$

Assumption 1 imposes smoothness on the optimal policy. Assumption 2 guarantees the existence of a local saddle point in the Lagrangian analysis. Assumption 3 refers to step sizes corresponding to policy updates and indicates that the update corresponding to $\{\alpha_{3,k}\}$ is on the fastest time-scale, the updates corresponding to $\{\alpha_{2,k}\}$ is on the intermediate time-scale, and the update corresponding to $\{\alpha_{1,k}\}$ is on the slowest time-scale. As this assumption refers to user-defined parameters, they can always be chosen to be satisfied.

To solve the CMDP, we employ the Lagrangian relaxation procedure (Bertsekas, 1999) to convert it to the following unconstrained problem:

$$\max_{\lambda \geq 0} \min_\theta \left( L(\theta, \lambda) \triangleq \mathcal{C}_{\pi_\theta}(x_0) + \lambda \left( \mathcal{D}_{\pi_\theta}(x_0) - d_0 \right) \right), \tag{11}$$

where $\lambda$ is the Lagrange multiplier. Notice that $L(\theta, \lambda)$ is a linear function in $\lambda$. Then, there exists a local saddle point $(\theta^*, \lambda^*)$ for the minimax optimization problem $\max_{\lambda \geq 0} \min_\theta L(\theta, \lambda)$, such that for some $r > 0$, $\forall \theta \in \mathbb{R}^\kappa \cap B_{\theta^*}(r)$, and $\forall \lambda \in [0, \lambda_{\max}]$, we have

$$L(\theta, \lambda^*) \geq L(\theta^*, \lambda^*) \geq L(\theta^*, \lambda), \tag{12}$$

where $B_{\theta^*}(r)$ is a hyper-dimensional ball centered at $\theta^*$ with radius $r > 0$.

In the following, we present a policy gradient (PG) algorithm and an actor-critic (AC) algorithm. While the PG algorithm updates its parameters after observing several trajectories, the AC algorithm is incremental and updates its parameters at each time-step.

We now present a policy gradient algorithm to solve the optimization problem (11). The idea of the algorithm is to descend in $\theta$ and ascend in $\lambda$ using the gradients of $L(\theta, \lambda)$ w.r.t. $\theta$ and $\lambda$, i.e.,

$$\nabla_\theta L(\theta, \lambda) = \nabla_\theta \left( \mathcal{C}_{\pi_\theta}(x_0) + \lambda \mathcal{D}_{\pi_\theta}(x_0) \right), \qquad \nabla_\lambda L(\theta, \lambda) = \mathcal{D}_{\pi_\theta}(x_0) - d_0. \tag{13}$$

The unit of observation in this algorithm is a system trajectory generated by following the current policy $\pi_{\theta_k}$. At each iteration, the algorithm generates $N$ trajectories by following the current policy $\pi_{\theta_k}$, uses them to estimate the gradients in (13), and then uses these estimates to update the parameters $\theta, \lambda$. Let $\xi = \{x_0, a_0, c_0, x_1, a_1, c_1, \ldots, x_{T-1}, a_{T-1}, c_{T-1}, x_T\}$ be a trajectory generated by following the policy $\theta$, where $x_T = x_{\text{Tar}}$ is the target state of the system and $T$ is the (random) stopping time. The cost, constraint cost, and probability of $\xi$ are defined as $\mathcal{C}(\xi) = \sum_{k=0}^{T-1} \gamma^k c(x_k, a_k)$, $\mathcal{D}(\xi) = \sum_{k=0}^{T-1} \gamma^k d(x_k)$, and $\mathbb{P}_\theta(\xi) = P_0(x_0) \prod_{k=0}^{T-1} \pi_\theta(a_k|x_k) P(x_{k+1}|x_k, a_k)$, respectively. Based on the definition of $\mathbb{P}_\theta(\xi)$, one obtains $\nabla_\theta \log \mathbb{P}_\theta(\xi) = \sum_{k=0}^{T-1} \nabla_\theta \log \pi_\theta(a_k|x_k)$.

Algorithm 1 contains the pseudo-code of our proposed PG algorithm. What appears inside the parentheses on the right-hand-side of the update equations are the estimates of the gradients of $L(\theta, \lambda)$ w.r.t. $\theta, \lambda$ (estimates of the expressions in (13)). Gradient estimates of the Lagrangian function are given by

$$\nabla_\theta L(\theta, \lambda) = \sum_\xi \mathbb{P}_\theta(\xi) \cdot \nabla_\theta \log \mathbb{P}_\theta(\xi) \left( \mathcal{C}_{\pi_\theta}(\xi) + \lambda \mathcal{D}_{\pi_\theta}(\xi) \right), \, \nabla_\lambda L(\theta, \lambda) = -d_0 + \sum_\xi \mathbb{P}_\theta(\xi) \cdot \mathcal{D}(\xi),$$

---

**Algorithm 1** Lagrangian Trajectory-based Policy Gradient Algorithm

---

**Input:** parameterized policy $\pi(\cdot|\cdot;\theta)$
**Initialization:** policy parameter $\theta = \theta_0$, and the Lagrangian parameter $\lambda = \lambda_0$
**for** $i = 0, 1, 2, \ldots$ **do**
  **for** $j = 1, 2, \ldots$ **do**
    Generate $N$ trajectories $\{\xi_{j,i}\}_{j=1}^N$ by starting at $x_0$ and following the policy $\theta_i$.
  **end for**

  $\theta$ **Update:** $\quad \theta_{i+1} = \theta_i - \alpha_{2,i} \dfrac{1}{N} \displaystyle\sum_{j=1}^N \nabla_\theta \log \mathbb{P}_\theta(\xi_{j,i})|_{\theta=\theta_i} \left( \mathcal{C}(\xi_{j,i}) + \lambda_i \mathcal{D}(\xi_{j,i}) \right)$

  $\lambda$ **Update:** $\quad \lambda_{i+1} = \Gamma_\Lambda \left[ \lambda_i + \alpha_{1,i} \left( -d_0 + \dfrac{1}{N} \displaystyle\sum_{j=1}^N \mathcal{D}(\xi_{j,i}) \right) \right]$

**end for**

---

where the likelihood gradient is

$$
\nabla_\theta \log \mathbb{P}_\theta(\xi) = \nabla_\theta \left\{ \sum_{k=0}^{T-1} \log P(x_{k+1}|x_k, a_k) + \log \pi_\theta(a_k|x_k) + \log \mathbf{1}\{x_0 = x^0\} \right\}
$$

$$
= \sum_{k=0}^{T-1} \nabla_\theta \log \pi_\theta(a_k|x_k) = \sum_{k=0}^{T-1} \frac{1}{\pi_\theta(a_k|x_k)} \nabla_\theta \pi_\theta(a_k|x_k).
$$

In Algorithm 1, $\Gamma_\Lambda$ is a projection operator to $[0, \lambda_{\max}]$, i.e., $\Gamma_\Lambda(\lambda) = \arg\min_{\hat\lambda \in [0, \lambda_{\max}]} \|\lambda - \hat\lambda\|_2^2$, which ensures the convergence of the algorithm. Recall from Assumption 3 that the step-size schedules satisfy the standard conditions for stochastic approximation algorithms, and ensure that the policy parameter $\theta$ update is on the fast time-scale $\{\alpha_{2,i}\}$, and the Lagrange multiplier $\lambda$ update is on the slow time-scale $\{\alpha_{1,i}\}$. This results in a two time-scale stochastic approximation algorithm that has been shown to converge to a (local) saddle point of the objective function $L(\theta, \lambda)$. This convergence proof makes use of standard results in stochastic approximation theory, because in the limit when the step-size is sufficiently small, analyzing the convergence of PG is equivalent to analyzing the stability of an ordinary differential equation (ODE) w.r.t. its equilibrium point.

In PG, the unit of observation is a system trajectory. This may result in high variance for the gradient estimates, especially when the length of the trajectories is long. To address this issue, we propose two actor-critic algorithms that use value function approximation in the gradient estimates and update the parameters incrementally (after each state-action transition). We present two actor-critic algorithms for optimizing (11). These algorithms are still based on the above gradient estimates. Algorithm 2 contains the pseudo-code of these algorithms. The projection operator $\Gamma_\Lambda$ is necessary to ensure the convergence of the algorithms. Recall from Assumption 3 that the step-size schedules satisfy the standard conditions for stochastic approximation algorithms, and ensure that the critic update is on the fastest time-scale $\{\alpha_{3,k}\}$, the policy update $\{\alpha_{2,k}\}$ is on the intermediate timescale, and finally the Lagrange multiplier update is on the slowest time-scale $\{\alpha_{1,k}\}$. This results in three time-scale stochastic approximation algorithms.

Using the PG theorem from (Sutton et al., 2000), one can show that

$$
\nabla_\theta L(\theta, \lambda) = \nabla_\theta V_\theta(x_0) = \frac{1}{1-\gamma} \sum_{x,a} \mu_\theta(x, a|x_0) \, \nabla \log \pi_\theta(a|x) \, Q_\theta(x, a), \tag{20}
$$

where $\mu_\theta$ is the discounted visiting distribution and $Q_\theta$ is the action-value function of policy $\theta$. We can show that $\frac{1}{1-\gamma} \nabla \log \pi_\theta(a_k|x_k) \cdot \delta_k$ is an unbiased estimate of $\nabla_\theta L(\theta, \lambda)$, where

$$
\delta_k = c_\lambda(x_k, a_k) + \gamma \hat{V}_\theta(x_{k+1}) - \hat{V}_\theta(x_k)
$$

is the temporal-difference (TD) error, and $\hat{V}_\theta$ is an estimator of the value function $V_\theta$.

Traditionally, for convergence guarantees in actor-critic algorithms, the critic uses linear approximation for the value function $V_\theta(x) \approx v^\top \psi(x) = \hat{V}_{\theta,v}(x)$, where the feature vector $\psi(\cdot)$ belongs to a low-dimensional space $\mathbb{R}^{\kappa_2}$. The linear approximation $\hat{V}_{\theta,v}$ belongs to a low-dimensional subspace

---

**Algorithm 2** Lagrangian Actor-Critic Algorithm

---

**Input:** Parameterized policy $\pi(\cdot|\cdot;\theta)$ and value function feature vector $\phi(\cdot)$
**Initialization:** policy parameters $\theta = \theta_0$; Lagrangian parameter $\lambda = \lambda_0$; value function weight $v = v_0$
**while** TRUE **do**
    **for** $k = 0, 1, 2, \ldots$ **do**
        Sample $a_k \sim \pi(\cdot|x_k;\theta_k)$; $c_{\lambda_k}(x_k, a_k) = c(x_k, a_k) + \lambda_k d(x_k)$; $x_{k+1} \sim P(\cdot|x_k, a_k)$;
        **// AC Algorithm:**

$$\textbf{TD Error:} \quad \delta_k(v_k) = c_{\lambda_k}(x_k, a_k) + \gamma \hat{V}_{\phi_k}(x_{k+1}) - \hat{V}_{\phi_k}(x_k) \tag{14}$$

$$\textbf{Critic Update:} \quad v_{k+1} = v_k + \zeta_3(k)\delta_k(v_k)\psi(x_k) \tag{15}$$

$$\boldsymbol{\theta} \textbf{ Update:} \quad \theta_{k+1} = \theta_k - \zeta_2(k)\nabla_\theta \log \pi_\theta(a_k|x_k) \cdot \delta_k(v_k)/1 - \gamma \tag{16}$$

$$\boldsymbol{\lambda} \textbf{ Update:} \quad \lambda_{k+1} = \Gamma_\Lambda\Big(\lambda_k + \zeta_1(k)\big(-d_0 + \frac{1}{N}\sum_{j=1}^{N}\mathcal{D}(\xi_{j,i})\big)\Big) \tag{17}$$

        **// NAC Algorithm:**

$$\textbf{Critic Update:} \quad w_{k+1} = \Big(I - \zeta_3(k)\nabla_\theta \log \pi_\theta(a_k|x_k)|_{\theta=\theta_k}\big(\nabla_\theta \log \pi_\theta(a_k|x_k)|_{\theta=\theta_k}\big)^\top\Big)w_k$$
$$+ \zeta_3(k)\delta_k(v_k)\nabla_\theta \log \pi_\theta(a_k|x_k)|_{\theta=\theta_k} \tag{18}$$

$$\boldsymbol{\theta} \textbf{ Update:} \quad \theta_{k+1} = \theta_k - \zeta_2(k)w_k/1 - \gamma \tag{19}$$

$$\textbf{Other Updates:} \quad \text{Follow from Eqs. 14, 15, and 17.}$$

    **end for**
**end while**

---

$S_V = \{\Psi v | v \in \mathbb{R}^{\kappa_2}\}$, where $\Psi$ is a short-hand notation for the set of features, i.e., $\Psi(x) = \psi^\top(x)$. Recently with the advances in deep neural networks, it has become increasingly popular to model the critic with a deep neural network, based on the objective function of minimizing the MSE of Bellman residual w.r.t. $V_\theta$ or $Q_\theta$ (Mnih et al., 2013).

## B   THE LYAPUNOV APPROACH TO SOLVE CMDPs

In this section, we revisit the *Lyapunov approach* to solving CMDPs that was proposed by (Chow et al., 2018) and report the mathematical results that are important in developing our safe policy optimization algorithms. To start, without loss of generality, we assume that we have access to a *baseline* feasible policy of (1), $\pi_B$; i.e., $\pi_B$ satisfies $\mathcal{D}_{\pi_B}(x_0) \leq d_0$. We define a set of Lyapunov functions w.r.t. initial state $x_0 \in \mathcal{X}$ and constraint threshold $d_0$ as

$$\mathcal{L}_{\pi_B}(x_0, d_0) = \{L : \mathcal{X} \to \mathbb{R}_{\geq 0} : T_{\pi_B, d}[L](x) \leq L(x), \forall x \in \mathcal{X}; \; L(x_0) \leq d_0\},$$

and call the constraints in this feasibility set the *Lyapunov constraints*. For any arbitrary Lyapunov function $L \in \mathcal{L}_{\pi_B}(x_0, d_0)$, we denote by

$$\mathcal{F}_L(x) = \left\{\pi(\cdot|x) \in \Delta : T_{\pi, d}[L](x) \leq L(x)\right\},$$

the set of $L$-induced Markov stationary policies. Since $T_{\pi, d}$ is a contraction mapping (Bertsekas, 2005), any $L$-induced policy $\pi$ has the property $\mathcal{D}_\pi(x) = \lim_{k\to\infty} T_{\pi, d}^k[L](x) \leq L(x)$, $\forall x \in \mathcal{X}$. Together with the property that $L(x_0) \leq d_0$, they imply that any $L$-induced policy is a feasible policy of (1). However, in general, the set $\mathcal{F}_L(x)$ does not necessarily contain an optimal policy of (1), and thus, it is necessary to design a Lyapunov function (w.r.t. a baseline policy $\pi_B$) that provides this guarantee. In other words, the main goal is to construct a Lyapunov function $L \in \mathcal{L}_{\pi_B}(x_0, d_0)$ such that

$$L(x) \geq T_{\pi^*, d}[L](x), \qquad L(x_0) \leq d_0. \tag{21}$$

Chow et al. (Chow et al., 2018) show in their Theorem 1 that **1)** without loss of optimality, the Lyapunov function can be expressed as

$$L_\epsilon(x) := \mathbb{E}\left[\sum_{t=0}^\infty \gamma^t (d(x_t) + \epsilon(x_t)) \mid \pi_B, x\right],$$

where $\epsilon(x) \geq 0$ is some auxiliary constraint cost uniformly upper-bounded by

$$\epsilon^*(x) := 2D_{\max} D_{TV}(\pi^* || \pi_B)(x)/(1 - \gamma),$$

and **2)** if the baseline policy $\pi_B$ satisfies the condition

$$\max_{x \in \mathcal{X}} \epsilon^*(x) \leq D_{\max} \cdot \min \left\{(1 - \gamma)\frac{d_0 - \mathcal{D}_{\pi_B}(x_0)}{D_{\max}}, \; \frac{D_{\max} - (1 - \gamma)\overline{\mathcal{D}}}{D_{\max} + (1 - \gamma)\overline{\mathcal{D}}}\right\},$$

where $\overline{\mathcal{D}} = \max_{x \in \mathcal{X}} \max_\pi \mathcal{D}_\pi(x)$ is the maximum constraint cost, then the Lyapunov function candidate $L_{\epsilon^*}$ also satisfies the properties of (21), and thus, its induced feasible policy set $\mathcal{F}_{L_{\epsilon^*}}$ contains an optimal policy. Furthermore, suppose that the distance between the baseline and optimal policies can be estimated efficiently. Using the set of $L_{\epsilon^*}$-induced feasible policies and noting that the *safe* Bellman operator $T[V](x) = \min_{\pi \in \mathcal{F}_{L_{\epsilon^*}}(x)} T_{\pi, c}[V](x)$ is monotonic and contractive, one can show that $T[V](x) = V(x)$, $\forall x \in \mathcal{X}$, has a unique fixed point $V^*$, such that $V^*(x_0)$ is a solution of (1) and an optimal policy can be constructed via greedification, i.e., $\pi^*(\cdot|x) \in \arg\min_{\pi \in \mathcal{F}_{L_{\epsilon^*}}(x)} T_{\pi, c}[V^*](x)$. This shows that under the above assumption, (1) can be solved using standard dynamic programming (DP) algorithms. While this result connects CMDP with Bellman's principle of optimality, verifying whether $\pi_B$ satisfies this assumption is challenging when a good estimate of $D_{TV}(\pi^* || \pi_B)$ is not available. To address this issue, Chow et al. (Chow et al., 2018) propose to approximate $\epsilon^*$ with an auxiliary constraint cost $\widetilde{\epsilon}$, which is the *largest* auxiliary cost satisfying the Lyapunov condition $L_{\widetilde{\epsilon}}(x) \geq T_{\pi_B, d}[L_{\widetilde{\epsilon}}](x)$, $\forall x \in \mathcal{X}$, and the safety condition $L_{\widetilde{\epsilon}}(x_0) \leq d_0$. The intuition here is that the larger $\widetilde{\epsilon}$, the larger the set of policies $\mathcal{F}_{L_{\widetilde{\epsilon}}}$. Thus, by choosing the largest such auxiliary cost, we hope to have a better chance of including the optimal policy $\pi^*$ in the set of feasible policies. Specifically, $\widetilde{\epsilon}$ is computed by solving the following linear program (LP):

$$\widetilde{\epsilon} \in \arg\max_{\epsilon : \mathcal{X} \to \mathbb{R}_{\geq 0}} \left\{\sum_{x \in \mathcal{X}} \epsilon(x) \; : \; d_0 - \mathcal{D}_{\pi_B}(x_0) \geq \mathbf{1}(x_0)^\top \left(I - \gamma\{P(x'|x, \pi_B(x))\}_{x, x' \in \mathcal{X}}\right)^{-1} \epsilon\right\}, \tag{22}$$

where $\mathbf{1}(x_0)$ represents a one-hot vector in which the non-zero element is located at $x = x_0$. When $\pi_B$ is a feasible policy, this problem has a non-empty solution. Furthermore, according to the derivations in (Chow et al., 2018), the maximizer of (22) has the following form:

$$\widetilde{\epsilon}(x) = \frac{(d_0 - \mathcal{D}_{\pi_B}(x_0)) \cdot \mathbf{1}\{x = \underline{x}\}}{\mathbb{E}\left[\sum_{t=0}^\infty \gamma^t \mathbf{1}\{x_t = \underline{x}\} \mid x_0, \pi_B\right]} \geq 0,$$

---

**Algorithm 3** Safe Policy Iteration (SPI)

---

**Input:** Initial feasible policy $\pi_0$;
**for** $k = 0, 1, 2, \ldots$ **do**
    **Step 0:** With $\pi_b = \pi_k$, evaluate the Lyapunov function $L_{\epsilon_k}$, where $\epsilon_k$ is a solution of (22)
    **Step 1:** Evaluate the cost value function $V_{\pi_k}(x) = \mathcal{C}_{\pi_k}(x)$; Then update the policy by solving the following
    problem: $\pi_{k+1}(\cdot|x) \in \mathrm{argmin}_{\pi \in \mathcal{F}_{L_{\epsilon_k}}(x)} T_{\pi,c}[V_{\pi_k}](x), \forall x \in \mathcal{X}$
**end for**
**Return** Final policy $\pi_{k^*}$

---

where $\underline{x} \in \arg\min_{x \in \mathcal{X}} \mathbb{E}\left[\sum_{t=0}^{\infty} \gamma^t \mathbf{1}\{x_t = x\} \mid x_0, \pi_B\right]$. They also show that by further restricting $\widetilde{\epsilon}(x)$ to be a constant function, the maximizer is given by

$$\widetilde{\epsilon}(x) = (1 - \gamma) \cdot (d_0 - \mathcal{D}_{\pi_B}(x_0)), \ \forall x \in \mathcal{X}.$$

Using the construction of the Lyapunov function $L_{\widetilde{\epsilon}}$, (Chow et al., 2018) propose the safe policy itera-tion (SPI) algorithm (see Algorithm 3) in which the Lyapunov function is updated via *bootstrapping*, i.e., at each iteration $L_{\widetilde{\epsilon}}$ is recomputed using (22) w.r.t. the current baseline policy. At each iteration $k$, this algorithm has the following properties: **1)** *Consistent Feasibility*, i.e., if the current policy $\pi_k$ is feasible, then $\pi_{k+1}$ is also feasible; **2)** *Monotonic Policy Improvement*, i.e., $\mathcal{C}_{\pi_{k+1}}(x) \leq \mathcal{C}_{\pi_k}(x)$ for any $x \in \mathcal{X}$; and **3)** *Asymptotic Convergence*. Despite all these nice properties, SPI is still a value-function-based algorithm, and thus, it is not straightforward to use it in continuous action problems. The main reason is that the greedification step becomes an optimization problem over the continuous set of actions that is not necessarily easy to solve. In Section 3, we show how we use SPI and its nice properties to develop safe policy optimization algorithms that can handle continuous action problems. Our algorithms can be thought as combinations of DDPG or PPO (or any other on-policy or off-policy policy optimization algorithm) with a SPI-inspired critic that evaluates the policy and computes its corresponding Lyapunov function. The computed Lyapunov function is then used to guarantee safe policy update, i.e., the new policy is selected from a restricted set of safe policies defined by the Lyapunov function of the current policy.

## C  Technical Details of the Safe Policy Gradient Algorithms

In this section, we first provide the details of the derivation of the $\theta$-projection and $a$-projection procedures described in Section 3, and then provide the pseudo-codes of our safe PG algorithms.

### C.1  Derivation of $\theta$-projection in Lyapunov-based Safe PG

To derive our $\theta$-projection algorithms, we first consider the original Lyapunov constraint in (3) that is given by

$$\int_{a\in\mathcal{A}} \left(\pi_\theta(a|x) - \pi_B(a|x)\right) Q_{L_{\pi_B}}(x,a) \, da \leq \widetilde{\epsilon}(x), \quad \forall x \in \mathcal{X},$$

where the baseline policy is parameterized as $\pi_B = \pi_{\theta_B}$. Using the first-order Taylor series expansion w.r.t. $\theta = \theta_B$, at any arbitrary $x \in \mathcal{X}$, the term $\mathbb{E}_{a\sim\pi_\theta}\left[Q_{L_{\theta_B}}(x,a)\right] = \int_{a\in\mathcal{A}} \pi_\theta(a|x) \, Q_{L_{\pi_B}}(x,a) \, da$ on left-hand-side of the above inequality can be written as

$$\mathbb{E}_{a\sim\pi_\theta}\left[Q_{L_{\theta_B}}(x,a)\right] = \mathbb{E}_{a\sim\pi_{\theta_B}}\left[Q_{L_{\theta_B}}(x,a)\right] + \left\langle (\theta-\theta_B), \nabla_\theta \mathbb{E}_{a\sim\pi_\theta}\left[Q_{L_{\theta_B}}(x,a)\right] |_{\theta=\theta_B} \right\rangle + O(\|\theta-\theta_B\|^2),$$

which implies that

$$\int_{a\in\mathcal{A}} \left(\pi_\theta(a|x) - \pi_B(a|x)\right) Q_{L_{\pi_B}}(x,a) \, da = \left\langle (\theta-\theta_B), \nabla_\theta \mathbb{E}_{a\sim\pi_\theta}\left[Q_{L_{\theta_B}}(x,a)\right] |_{\theta=\theta_B} \right\rangle + O(\|\theta-\theta_B\|^2).$$

Note that the objective function of the constrained minimization problem in (4) contains a regularization term: $\frac{\beta}{2}\left\langle (\theta - \theta_B), \nabla_\theta^2 \overline{D}_{\mathrm{KL}}(\theta, \theta_B) |_{\theta=\theta_B} (\theta - \theta_B) \right\rangle$ that controls the distance $\|\theta - \theta_B\|$ to be small. For most practical purposes, here one can assume the higher-order term $O(\|\theta - \theta_B\|^2)$ to be much smaller than the first-order term $\left\langle (\theta - \theta_B), \nabla_\theta \mathbb{E}_{a\sim\pi_\theta}\left[Q_{L_{\theta_B}}(x,a)\right] |_{\theta=\theta_B} \right\rangle$. Therefore, one can approximate the original Lyapunov constraint in (3) with the following constraint:

$$\left\langle (\theta - \theta_B), \nabla_\theta \mathbb{E}_{a\sim\pi_\theta}\left[Q_{L_{\theta_B}}(x,a)\right] |_{\theta=\theta_B} \right\rangle \leq \widetilde{\epsilon}(x), \quad \forall x \in \mathcal{X}.$$

Furthermore, following the same line of arguments used in TRPO (to transform the $\max D_{\mathrm{KL}}$ constraint to an average $\overline{D}_{\mathrm{KL}}$ constraint, see Eq. 12 in (Schulman et al., 2015a)), a more numerically stable way is to *approximate* the Lyapunov constraint using the average constraint surrogate, i.e.,

$$\left\langle (\theta - \theta_B), \frac{1}{M}\sum_{i=1}^{M} \nabla_\theta \mathbb{E}_{a\sim\pi_\theta}\left[Q_{L_{\theta_B}}(x_i,a)\right] |_{\theta=\theta_B} \right\rangle \leq \frac{1}{M}\sum_{i=1}^{M} \widetilde{\epsilon}(x_i).$$

Now consider the special case when auxiliary constraint surrogate is chosen as a constant, i.e., $\widetilde{\epsilon} = (1-\gamma)\left(d_0 - \mathcal{D}_{\pi_{\theta_B}}(x_0)\right)$. The justification of such choice comes from analyzing the solution of optimization problem (22). Then, one can write the Lyapunov action-value function $Q_{L_{\theta_B}}(x,a)$ as

$$Q_{L_{\theta_B}}(x,a) = \mathbb{E}\left[\sum_{t=0}^{\infty} \gamma^t d(x_t)|\pi_B, x_0 = x, a_0 = a\right] + \frac{\widetilde{\epsilon}}{1-\gamma}.$$

Since the second term is independent of $\theta$, for any state $x \in \mathcal{X}$, the gradient term $\nabla_\theta \mathbb{E}_{a\sim\pi_\theta}\left[Q_{L_{\theta_B}}(x,a)\right]$ can be simplified as

$$\nabla_\theta \mathbb{E}_{a\sim\pi_\theta}\left[Q_{L_{\theta_B}}(x,a)\right] = \int_a \pi_\theta(a|x) \, \nabla_\theta \log \pi_\theta(a|x) \, Q_{W_{\theta_B}}(x,a)da = \nabla_\theta \mathbb{E}_{a\sim\pi_\theta}\left[Q_{W_{\theta_B}}(x,a)\right],$$

where $W_{\theta_B}(x) = T_{\pi_B,d}[W_{\theta_B}](x)$ and $Q_{W_{\theta_B}}(x,a) = d(x) + \gamma\sum_{x'} P(x'|x,a)W_{\theta_B}(x')$ are the constraint value function and constraint state-action value function, respectively. The second equality is based on the standard log-likelihood gradient property in PG algorithms (Sutton et al., 2000).

Collectively, one can then re-write the Lyapunov average constraint surrogate as

$$\left\langle (\theta - \theta_B), \frac{1}{M}\sum_{i=1}^{M} \nabla_\theta \mathbb{E}_{a\sim\pi_\theta}\left[Q_{W_{\theta_B}}(x_i,a)\right] |_{\theta=\theta_B} \right\rangle \leq \widetilde{\epsilon},$$

where $\widetilde{\epsilon}$ is the auxiliary constraint cost defined specifically by the Lyapunov-based approach, to guarantee constraint satisfaction. By expanding the auxiliary constraint cost $\widetilde{\epsilon}$ on the right-hand-side, the above constraint is equivalent to the constraint used in CPO, i.e.,

$$\mathcal{D}_{\pi_{\theta_B}}(x_0) + \frac{1}{1-\gamma}\langle (\theta - \theta_B), \frac{1}{M}\sum_{i=1}^{M}\nabla_\theta \mathbb{E}_{a\sim\pi_\theta}[Q_{W_{\theta_B}}(x_i,a)]|_{\theta=\theta_B}\rangle \leq d_0.$$

## C.2 Derivations of $a$-projection in Lyapunov-based Safe PG

For any arbitrary state $x \in \mathcal{X}$, consider the following constraint in the safety-layer projection problem given in (6):

$$Q_{L_{\pi_B}}(x, a) - Q_{L_{\pi_B}}(x, \pi_B(x)) \leq \widetilde{\epsilon}(x).$$

Using first-order Taylor series expansion of the Lyapunov state-action value function $Q_{L_{\pi_B}}(x, a)$ w.r.t. action $a = \pi_B(x)$, the Lyapunov value function $Q_{L_{\pi_B}}(x, a)$ can be re-written as

$$Q_{L_{\pi_B}}(x, a) = Q_{L_{\pi_B}}(x, \pi_B(x)) + (a - \pi_B(x))^\top g_{L_{\pi_B}}(x) + O(\|a - \pi_B(x)\|^2).$$

Note that the objective function of the action-projection problem in (7) contains a regularization term $\frac{\eta(x)}{2} \|a - \pi_B(x)\|^2$ that controls the distance $\|a - \pi_B(x)\|$ to be small. For most practical purposes, here one can assume the higher-order term $O(\|a - \pi_B(x)\|^2)$ to be much smaller than the first-order term $(a - \pi_B(x))^\top g_{L_{\pi_B}}(x)$. Therefore, one can approximate the original action-based Lyapunov constraint in (6) with the constraint $\left( a - \pi_B(x) \right)^\top g_{L_{\pi_B}}(x) \leq \widetilde{\epsilon}(x)$ that is the constraint in (7). Similar to the analysis of the $\theta$-projection approach, if the auxiliary cost $\widetilde{\epsilon}$ is state-independent, the action-gradient term $g_{L_{\pi_B}}(x)$ is equal to the gradient of the constraint action-value function $\nabla_a Q_{W_{\theta_B}}(x, a) \mid_{a=\pi_B(x)}$, where $Q_{W_{\theta_B}}$ is the state-action constraint value function w.r.t. the baseline policy. The rest of the proof follows the results from Proposition 1 in (Dalal et al., 2018). This completes the derivations of the $a$-projection approach.

## C.3 Pseudo-codes of Our Safe PG Algorithms

Algorithms 4 and 5 contain the pseudo-code of our safe Lyapunov-based policy gradient (PG) algorithms with $\theta$-projection and $a$-projection, respectively.

## C.4 Practical Implementation of Our Safe PG Algorithms

Due to function approximation errors, even with the Lyapunov constraints, in practice a safe PG algorithm may take a bad step and produce an infeasible policy update and cannot automatically recover from such a bad step. To address this issue, similar to (Achiam et al., 2017), we propose the following *safeguard* policy update rule to decrease the constraint cost: $\theta_{k+1} = \theta_k - \alpha_{\text{sg},k} \nabla_\theta \mathcal{D}_{\pi_\theta}(x_0)_{\theta=\theta_k}$, where $\alpha_{\text{sg},k}$ is the learning rate for the safeguard update. If $\alpha_{\text{sg},k} >> \alpha_k$ (learning rate of PG), then with the safeguard update, $\theta$ will quickly recover from the bad step, however, it might be overly conservative. This approach is principled because as soon as $\pi_{\theta_k}$ is unsafe/infeasible w.r.t. the CMDP constraints, the algorithm uses a limiting search direction. One can directly extend this safeguard update to the multiple-constraint scenario by doing gradient descent over the constraint that has the worst violation.

Another remedy to reduce the chance of constraint violation is to do *constraint tightening* on the constraint cost threshold. Specifically, instead of $d_0$, one may pose the constraint based on $d_0 \cdot (1 - \delta)$, where $\delta \in (0, 1)$ is the factor of safety for providing additional buffer to constraint violation. Additional techniques in cost-shaping have been proposed in (Achiam et al., 2017) to smooth out the sparse constraint costs. While these techniques can further ensure safety, construction of the cost-shaping term requires knowledge of the environment, which makes the safe PG algorithms more complicated.

---

**Algorithm 4** Lyapunov-based Safe PG with $\theta$-projection (SDDPG and SPPO)

---

**Input:** Initial feasible policy $\pi_0$;

**for** $k = 0, 1, 2, \ldots$ **do**

**Step 0:** With $\pi_b = \pi_{\theta_k}$, generate $N$ trajectories $\{\xi_{j,k}\}_{j=1}^N$ of $T$ steps by starting at $x_0$ and following the policy $\theta_k$

**Step 1:** Using the trajectories $\{\xi_{j,k}\}_{j=1}^N$, estimate the critic $Q_\theta(x, a)$ and the constraint critic $Q_{D,\theta}(x, a)$;
- For DDPG, these functions are trained by minimizing the MSE of Bellman residual, and one can also use off-policy samples from replay buffer (Schaul et al., 2015);
- For PPO these functions can be estimated by the generalized advantage function technique from Schulman et al. (2015b)

**Step 2:** Based on the closed form solution of a QP problem with an LP constraint in Section 10.2 of Achiam et al. (2017), calculate $\lambda_k^*$ with the following formula:

$$\lambda_k^* = \left( \frac{-\beta_k \widetilde{\epsilon} - \left( \nabla_\theta Q_\theta(\bar{x}, \bar{a}) \mid_{\theta=\theta_k} \right)^\top H(\theta_k)^{-1} \nabla_\theta Q_{D,\theta}(\bar{x}, \bar{a}) \mid_{\theta=\theta_k}}{\left( \nabla_\theta Q_{D,\theta}(\bar{x}, \bar{a}) \mid_{\theta=\theta_k} \right)^\top H(\theta_k)^{-1} \nabla_\theta Q_{D,\theta}(\bar{x}, \bar{a}) \mid_{\theta=\theta_k}} \right)_+ ,$$

where

$$\nabla_\theta Q_\theta(\bar{x}, \bar{a}) = \frac{1}{N} \sum_{x,a \in \xi_{j,k}, 1 \leq j \leq N} \sum_{t=0}^{T-1} \gamma^t \nabla_\theta \log \pi_\theta(a|x) Q_\theta(x, a),$$

$$\nabla_\theta Q_{D,\theta}(\bar{x}, \bar{a}) = \frac{1}{N} \sum_{x,a \in \xi_{j,k}, 1 \leq j \leq N} \sum_{t=0}^{T-1} \gamma^t \nabla_\theta \log \pi_\theta(a|x) Q_\theta(x, a),$$

$\beta_k$ is the adaptive penalty weight of the $\overline{D}_{\mathrm{KL}}(\pi||\pi_{\theta_k})$ regularizer, and $H(\theta_k) = \nabla_\theta^2 \overline{D}_{\mathrm{KL}}(\pi||\pi_\theta) \mid_{\theta=\theta_k}$ is the Hessian of this term

**Step 3:** Update the policy parameter by following the objective gradient;
- For DDPG

$$\theta_{k+1} \leftarrow \theta_k - \alpha_k \cdot \frac{1}{N \cdot T} \sum_{x \in \xi_{j,k}, 1 \leq j \leq N} \nabla_\theta \pi_\theta(x) \mid_{\theta=\theta_k} \cdot \left( \nabla_a Q_{\theta_k}(x, a) + \lambda_k^* \nabla_a Q_{D,\theta_k}(x, a) \right) \mid_{a=\pi_{\theta_k}(x)}$$

- For PPO,

$$\theta_{k+1} \leftarrow \theta_k - \frac{\alpha_k}{N\beta_k} \left( H(\theta_k) \right)^{-1} \sum_{x_{j,t}, a_{j,t} \in \xi_{j,k}, 1 \leq j \leq N} \sum_{t=0}^{T-1} \gamma^t \cdot \nabla_\theta \log \pi_\theta(a_{j,t}|x_{j,t}) \mid_{\theta=\theta_k} \cdot$$
$$\left( Q_{\theta_k}(x_{j,t}, a_{j,t}) + \lambda_k^* Q_{D,\theta_k}(x_{j,t}, a_{j,t}) \right)$$

**Step 4:** At any given state $x \in \mathcal{X}$, compute the feasible action probability $a^*(x)$ via action projection in the safety layer, that takes inputs $\nabla_a Q_L(x, a) = \nabla_a Q_{D,\theta_k}(x, a)$ and $\epsilon(x) = (1 - \gamma)(d_0 - Q_{D,\theta_k}(x_0, \pi_k(x_0)))$, for any $a \in \mathcal{A}$.

**end for**

**Return** Final policy $\pi_{\theta_{k^*}}$,

---

## D    Experimental Setup in MuJoCo Tasks

Our experiments are performed on safety-augmented versions of standard MuJoCo domains (Todorov et al., 2012).

**HalfCheetah-Safe.** The agent is a the standard HalfCheetah (a 2-legged simulated robot rewarded for running at high speed) augmented with safety constraints. We choose the safety constraints to be defined on the speed limit. We constrain the speed to be less than 1, i.e., constraint cost is thus $\mathbf{1}[|v| > 1]$. Episodes are of length 200. The constraint threshold is 50.

**Point Circle.** This environment is taken from (Achiam et al., 2017). The agent is a point mass (controlled via a pivot). The agent is initialized at $(0, 0)$ and rewarded for moving counter-clockwise along a circle of radius 15 according to the reward $\frac{-dx \cdot y + dy \cdot x}{1 + |\sqrt{x^2 + y^2} - 15|}$, for position $x, y$ and velocity $dx, dy$. The safety constraint is defined as the agent staying in a position satisfying $|x| \leq 2.5$. The constraint cost is thus $\mathbf{1}[|x| > 2.5]$. Episodes are of length 65. The constraint threshold is 7.

---

**Algorithm 5** Lyapunov-based Safe PG (SDDPG and SPPO) with $a$-projection

---

**Input:** Initial feasible policy $\pi_0$;
**for** $k = 0, 1, 2, \ldots$ **do**
    **Step 0:** With $\pi_b = \pi_{\theta_k}$, generate $N$ trajectories $\{\xi_{j,k}\}_{j=1}^N$ of $T$ steps by starting at $x_0$ and following the policy $\theta_k$
    **Step 1:** Using the trajectories $\{\xi_{j,k}\}_{j=1}^N$, estimate the critic $Q_\theta(x, a)$ and the constraint critic $Q_{D,\theta}(x, a)$;
        • For DDPG, these functions are trained by minimizing the MSE of Bellman residual, and one can also use off-policy samples from replay buffer (Schaul et al., 2015);
        • For PPO these functions can be estimated by the generalized advantage function technique from Schulman et al. (2015b)
    **Step 2:** Update the policy parameter by following the objective gradient;
        • For DDPG

$$\theta_{k+1} \leftarrow \theta_k - \alpha_k \cdot \frac{1}{N \cdot T} \sum_{x \in \xi_{j,k}, 1 \le j \le N} \nabla_\theta \pi_\theta(x) \mid_{\theta=\theta_k} \cdot \nabla_a Q_{\theta_k}(x, a) \mid_{a=\pi_{\theta_k}(x)};$$

        • For PPO,

$$\theta_{k+1} \leftarrow \theta_k - \frac{\alpha_k}{N\beta_k} \left(H(\theta_k)\right)^{-1} \sum_{x_{j,t}, a_{j,t} \in \xi_{j,k}, 1 \le j \le N} \sum_{t=0}^{T-1} \gamma^t \cdot \nabla_\theta \log \pi_\theta(a_{j,t} | x_{j,t}) \mid_{\theta=\theta_k} \cdot Q_{\theta_k}(x_{j,t}, a_{j,t})$$

        where $\beta_k$ is the adaptive penalty weight of the $\overline{D}_{\text{KL}}(\pi || \pi_{\theta_k})$ regularizer, and $H(\theta_k) = \nabla_\theta^2 \overline{D}_{\text{KL}}(\pi || \pi_\theta) \mid_{\theta=\theta_k}$ is the Hessian of this term
    **Step 3:** At any given state $x \in \mathcal{X}$, compute the feasible action probability $a^*(x)$ via action projection in the safety layer, that takes inputs $\nabla_a Q_L(x, a) = \nabla_a Q_{D,\theta_k}(x, a)$ and $\epsilon(x) = (1 - \gamma)(d_0 - Q_{D,\theta_k}(x_0, \pi_k(x_0)))$, for any $a \in \mathcal{A}$.
**end for**
**Return** Final policy $\pi_{\theta_{k^*}}$,

---

**Point Gather.** This environment is taken from (Achiam et al., 2017). The agent is a point mass (controlled via a pivot) and the environment includes randomly positioned apples (2 apples) and bombs (8 bombs). The agent given a reward of 10 for each apple collected and a penalty of $-10$ for each bomb. The safety constraint is defined as the number of bombs collected during the episode. Episodes are of length 15. The constraint threshold is 4 for DDPG and 2 for PPO.

**Ant Gather.** This environment is the same as Point Circle, only with an Ant agent (quadrapedal simulated robot). Each episode is initialized with 8 apples and 8 bombs. The agent receives a reward of 10 for each apple collected, a penalty of $-20$ for each bomb collected, and a penalty of $-20$ if the episode terminates prematurely (because the Ant falls). Episodes are of length at most 500. The constraint threshold is 10 and 5 for DDPG and PPO, respectively.

Figure 7 shows the visualization of the above domains used in our experiments.

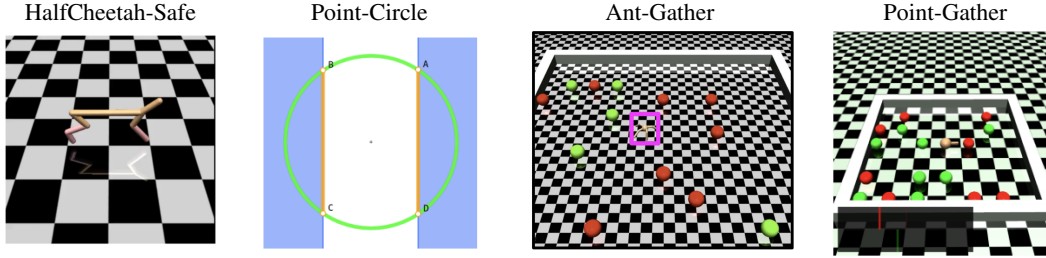

|  HalfCheetah-Safe | Point-Circle | Ant-Gather | Point-Gather |

Figure 7: The Robot Locomotion Control Tasks

In these experiments, there are three different agents: (1) a point-mass ($\mathcal{X} \subseteq \mathbb{R}^9$, $A \subseteq \mathbb{R}^2$); an ant quadruped robot ($\mathcal{X} \subseteq \mathbb{R}^{32}$, $A \subseteq \mathbb{R}^8$); and (3) a half-cheetah ($\mathcal{X} \subseteq \mathbb{R}^{18}$, $A \subseteq \mathbb{R}^6$). For all experiments, we use two neural networks with two hidden layers of size $(100, 50)$ and ReLU activation to model the mean and log-variance of the Gaussian actor policy, and two neural networks with two hidden layers of size $(200, 50)$ and tanh activation to model the critic and constraint critic. To build a low variance sample gradient estimate, we use GAE-$\lambda$ (Schulman et al., 2015b) to estimate

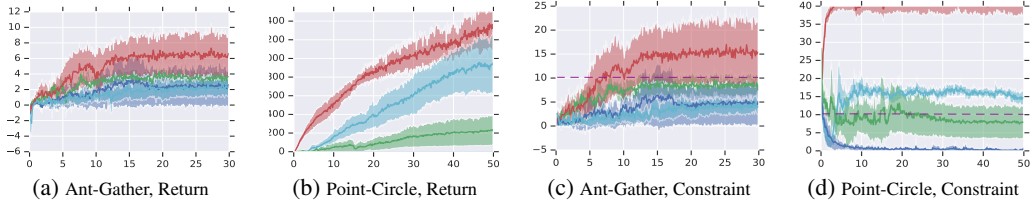

(a) Ant-Gather, Return    (b) Point-Circle, Return    (c) Ant-Gather, Constraint    (d) Point-Circle, Constraint

Figure 8: DDPG (red), DDPG-Lagrangian (cyan), SDDPG (blue), SDDPG $a$-projection (green) on Ant-Gather and Point-Circle. Ours SDDPG and SDDPG $a$-projection algorithms perform stable and safe learning, although the dynamics and cost functions are unknown, control actions are continuous, and deep function approximation is used. Unit of x-axis is in thousands of episodes. Shaded areas represent the 1-SD confidence intervals (over 10 random seeds). The dashed purple line in the two right figures represents the constraint limit.

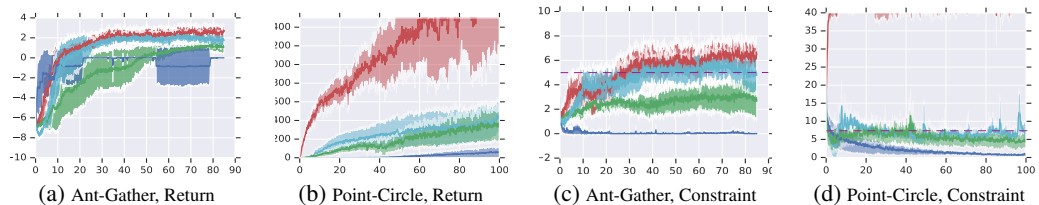

(a) Ant-Gather, Return    (b) Point-Circle, Return    (c) Ant-Gather, Constraint    (d) Point-Circle, Constraint

Figure 9: PPO (red), PPO-Lagrangian (cyan), SPPO (blue), SPPO $a$-projection (green) on Ant-Gather and Point-Circle. SPPO $a$-projection performs stable and safe learning, when the dynamics and cost functions are unknown, control actions are continuous, and deep function approximation is used.

the advantage and constraint advantage functions, with a hyper-parameter $\lambda \in (0, 1)$ optimized by grid-search.

On top of GAE-$\lambda$, in all experiments and for each algorithm (SDDPG, SPPO, SDDPG $a$-projection, SPPO $a$-projection, CPO, Lagrangian, and the unconstrained PG counterparts), we systematically explored different parameter settings by doing grid-search over the following factors: (i) learning rates in the actor-critic algorithm, (ii) batch size, (iii) regularization parameters of the policy relative entropy term, (iv) with-or-without natural policy gradient updates, (v) with-or-without the emergency safeguard PG updates (see Appendix C.4 for more details). Although each algorithm might have a different parameter setting that leads to the optimal performance in training, the results reported here are the best ones for each algorithm, chosen by the same criteria (which is based on the value of return plus certain degree of constraint satisfaction). To account for the variability during training, in each learning curve, a 1-SD confidence interval is also computed over 10 separate random runs (under the same parameter setting).

## D.1    MORE EXPLANATIONS ON MUJOCO RESULTS

In all numerical experiments and for each algorithm (SPPO $\theta$-projection, SDDPG $\theta$-projection, SPPO $a$-projection, SDDPG $a$-projection, CPO, Lagrangian, and the unconstrained PG counterparts), we systematically explored various hyper-parameter settings by doing grid-search over the following factors: (i) learning rates in the actor-critic algorithm, (ii) batch size, (iii) regularization parameters of the policy relative entropy term, (iv) with-or-without natural policy gradient updates, (v) with-or-without the emergency safeguard PG updates (see Appendix C.4 for more details). Although each algorithm might have a different parameter setting that leads to the optimal training performance, the results reported in the paper are the best ones for each algorithm, chosen by the same criteria (which is based on value of return + certain degree of constraint satisfaction).

In our experiments, we compare the two classes of safe RL algorithms, one derived from $\theta$-projection (constrained policy optimization) and one from the $a$-projection (safety layer), with the unconstrained and Lagrangian baselines in four problems: PointGather, AntGather, PointCircle, and HalfCheetah-Safe. We perform these experiments with both off-policy (DDPG) and on-policy (PPO) versions of the algorithms.

In PointCircle DDPG, although the Lagrangian algorithm significantly outperforms the safe RL algorithms in terms of return, it violates the constraint more often. The only experiment in which Lagrangian performs similarly to the safe algorithms in terms of both return and constraint violation is PointCircle PPO. In all other experiments that are performed in the HalfCheetahSafe, PointGather and

AntGather domains, either (i) the policy learned by Lagrangian has a significantly lower performance than that learned by one of the safe algorithms (see HalfCheetahSafe DDPG, PointGather DDPG, AntGather DDPG), or (ii) the Lagrangian method violates the constraint during training, while the safe algorithms do not (see HalfCheetahSafe PPO, PointGather PPO, AntGather PPO). This clearly illustrates the effectiveness of our Lyapunov-based safe RL algorithms, when compared to Lagrangian method.

# E  EXPERIMENTAL SETUP IN THE ROBOT NAVIGATION PROBLEM

Mapless navigation task is a continuous control task with a goal of navigating a robot to any arbitrary goal position collision-free and without memory of the workspace topology. The goal is usually within $5 - 10$ meters from the robot agent, but it is not visible to the agent before the task starts, due to both limited sensor range and the presence of obstacles that block a clear line of sight. The agent's observations, $\boldsymbol{x} = (\boldsymbol{g}, \dot{\boldsymbol{g}}, \boldsymbol{l}) \in \mathbb{R}^{68}$, consists of the relative goal position, the relative goal velocity, and the Lidar measurements. Relative goal position, $\boldsymbol{g}$, is the relative polar coordinates between the goal position and the current robot pose, and $\dot{\boldsymbol{g}}$ is the time derivative of $\boldsymbol{g}$, which indicates the speed of the robot navigating to the goal. This information is available from the robot's localization sensors. Vector $\boldsymbol{l}$ is the noisy Lidar input (Fig. 3a), which measures the nearest obstacle in a direction within a $220°$ field of view split in $64$ bins, up to $5$ meters in depth. The action is given by $\boldsymbol{a} \in \mathbb{R}^2$, which is linear and angular velocity vector at the robot's center of the mass. The transition probability $P : \mathcal{X} \times \mathcal{A} \to \mathcal{X}$ captures the noisy differential drive robot dynamics. Without knowing the full non-linear system dynamics, we here assume knowledge of a simplified blackbox kinematics simulator operating at 5Hz in which Gaussian noise, $\mathcal{N}(0, 0.1)$, is added to both the observations and actions in order to model the noise in sensing, dynamics, and action actuations in real-world. The objective of the P2P task is to navigate the robot to reach within 30 centimeters from any real-time goal. While the dynamics of this system is simpler than that of HalfCheetah. But unlike the MuJoCo tasks where the underlying dynamics are deterministic, in this robot experiment the sensor, localization, and dynamics noise paired with partial world observations and unexpected obstacles make this safe RL much more challenging. More descriptions about the indoor robot navigation problem and its implementation details can be found in Section 3 and 4 of (Chiang et al., 2019). Fetch robot weights 150 kilograms, and reaches maximum speed of 7 km/h making the collision force a safety paramount.

Here the CMDP is non-discounting and has a finite-horizon of $T = 100$. We reward the agent for reaching the goal, which translates to an immediate cost of $c(\boldsymbol{x}, \boldsymbol{a}) = \|g\|^2$, which measures the relative distance to goal. To measure the impact energy of obstacle collisions, we impose an immediate constraint cost of $d(\boldsymbol{x}, \boldsymbol{a}) = \|\dot{\boldsymbol{g}}\| \cdot \mathbf{1}\{\|\boldsymbol{l}\| \leq r_{\text{impact}}\}/T$, where $r_{\text{impact}}$ is the impact radius w.r.t. the Lidar depth signal, to account for the speed during collision, with a constraint threshold $d_0$ that characterizes the agent's maximum tolerable collision impact energy to any objects. (Here the total impact energy is proportional to the robot's speed during any collisions.) Under this CMDP framework (Fig. 3b), the main goal is to train a policy $\pi^*$ that drives the robot along the shortest path to the goal and to limit the average impact energy of obstacle collisions. Furthermore, due to limited data any intermediate point-to-point policy is deployed on the robot to collect more samples for further training, therefore guaranteeing safety during training is critical in this application.

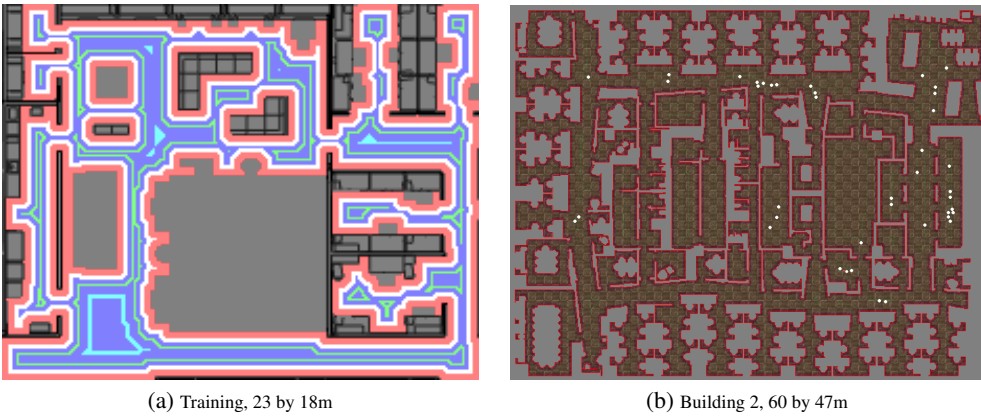

(a) Training, 23 by 18m  (b) Building 2, 60 by 47m

Figure 10: (a) Training and (b) evaluation environments, generated from real office building plans. The evaluation environment is an order of magnitude bigger.

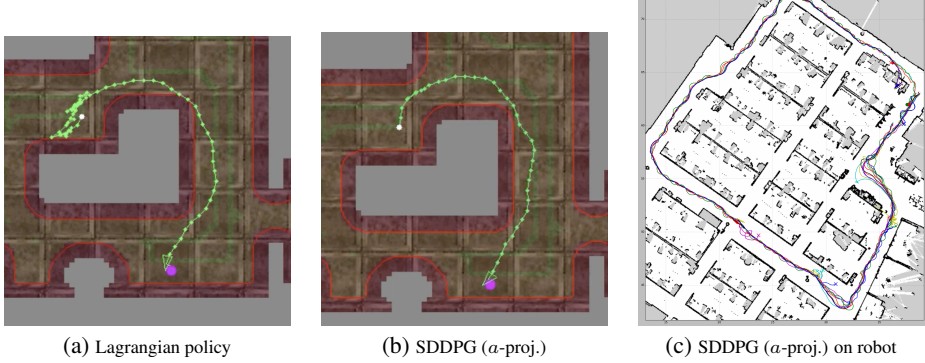

(a) Lagrangian policy       (b) SDDPG ($a$-proj.)       (c) SDDPG ($a$-proj.) on robot

Figure 11: Navigation routes of two policies on a similar setup (a) and (b). Log of on-robot experiments (c).

