# OpenReview forum: "Safe Policy Learning for Continuous Control"
_ICLR.cc/2020/Conference — Reject_

### Official Review · AnonReviewer1 · 2019-10-21
**Official Blind Review #1**

**Rating:** 6

**Review:**


Summary:

Authors propose ideas to perform safe RL in continuous actions domain with modifications to Policy Gradient (PG)  algorithms via either constraining the policy parameters or constraining the actions selected by PG with a surrogate/augmented state dependent objective. The paper is well motivated and the experiments (although I have some reservations about the setup) demonstrate efficacy of the proposed method.

Review:
--> Introduction
I do not agree with the statement that value function based algorithms are restricted to discrete action domains, especially when you rely on  “ignoring function approximation errors” for some of your claims.

Again in switching from value function to PG is true for traditional RL/Control theory but this is not valid here. ( your methods rely on Q(s, a) which is action-value function, or the constraint in equation 3 is integral of Q over all actions which would be value-function in traditional definition )
Note: this is explained very well towards the end in the Appendix B, but this is a review of the paper and not Appendix B or C.


--> Section 2
section 2.3 I would strongly advise the authors to rewrite this, this section reads like it was copied as is from the reference [Chow et al 2018].  especially the way Lyapunov function is defined. And the language and arguments are almost same. Some sentences cite the reference but conclusions drawn on these are not cited, are you claiming that these conclusions are original from this paper ?

It is not clear to me how the feasibility of initial pi_0 is ensured ? Did I miss this somewhere ?


→ Section 3

Section 3 is pleasant to read and very easy to understand, however, same cannot be said of the
section 3.1. I had to spend significant time reading 3.1 and I am still not sure I have understood it very well.

Experiments:

I don’t think halfCheetah-Safe is actually actually an useful experiment, Limiting the joint torques is perfectly understandable, just limiting speed and getting smooth motion could just be an artifact of the simulation environment. Are both constraints applied simultaneously (torque and speed) ? It is unclear from the text.

I am not sure CPO without linesearch is actually a fair comparison. Line search may actually deem of the actions unsafe and therefore I would presume original CPO do be less prone to constraint violation than the proposed modification in your experiments. Again PPO is more heuristic than TRPO which makes it hard to compare like for like. PPO might give higher rewards but constraint violations may increase as well. An important point for Safe-RL I feel.

Figure 6
Can you be more specific as what the figure 6 is showing ? Constraint ? is this constraint violation count? or cumulative sum of constraint slack over the whole trajectory ?


Not part of assessment :


Unclear Statements:

Page 7, DDPG Vs PPO: explain clearly what you mean by  “covariate shift” or remove the statement altogether.

Page 7, section 5 second paragraph, “The actions are the linear …. center of mass” I couldn’t understand this ? What do you mean by actions are velocity ?



Minor points (Language, Typos):

page 3, last paragraph,  Chow et al. is repeated, I can see why this happens there but suggest editing to avoid this. [This is also in intro paragraph, there it is just a typo and should be rectified]

Figure 6: Captions labels are incorrect.







**Experience Assessment:**

I have published one or two papers in this area.

**Review Assessment: Checking Correctness Of Derivations And Theory:**

I assessed the sensibility of the derivations and theory.

**Review Assessment: Checking Correctness Of Experiments:**

I carefully checked the experiments.

**Review Assessment: Thoroughness In Paper Reading:**

I read the paper thoroughly.

---

> ### Author Response · Authors · 2019-11-07
> **Thank you. Please find our response below.**
>
> We thank the reviewer for providing useful feedbacks. Please find the itemized feedback to your questions/comments below:
>
> Value-based algorithm in intro:
> We originally refer to the Lyapunov approach in Chow’18, which primarily relies on the discrete action space assumption (that is key for defining the LP formulation of the feasibility set F_L). It is true that value-based methods can also be used in continuous action RL, although it is less common than policy-based method because solving the max-Q problem can be computationally complex. To avoid these misnomers, we will add more descriptions in the introduction section regarding value-based RL for continuous control in the final paper.
>
> Section 2:
> We mostly adopt the descriptions of Lyapunov functions from Chow’18. We made it clear in the introduction section that using Lyapunov functions to guarantee safety in RL is not a novel contribution of this paper. To improve the readability of this section, we will rewrite the descriptions in this section and add the missing references in the final paper.
>
> pi_0:
> Existence of a feasible \pi_0 can be checked/ensured by minimizing the cumulative constraint cost function. If the corresponding optimal policy satisfies the constraint, one can simply use that as \pi_0. Otherwise, the problem has an empty feasibility set. This argument can be found in Chow’18, and we will add these descriptions about pi_0 in the final paper.
>
> Section 3.1:
> We will modify the flow of presentation and streamline the mathematical expressions to improve the readability of this section in the final paper.
>
> Figure 6:
> Figure 6a shows the average success rate over 100 tasks (randomly sampled start and goal robot positions). A task is considered successful if the robot reaches the goal, regardless of the constraint. This is difficult because the robot navigates without the map, relying only on its noisy sensors. Figure 6b shows the average cumulative cost over those 100 tasks. Specifically, for each task we report the constraint experienced over the entire trajectory, and average that over the 100 trials. We will clarify that in the caption of the Figure.
>
> Experiments:
> HalfCheetah Safe: In this experiment our constraint is on the speed of the Cheetah. Restricting the joint torque is also an alternative constraint that we tried (but we omit their numerical results for the sake of brevity). We only apply one of the constraints in each experiment. Empirically, policies trained in these two types of constraints have similar performance, while intuitively these two cases have different meanings (bounding the total speed versus bounding the total torque).
>
> Comparison with CPO without linesearch:
> Here we choose to compare the Lyapunov-based approaches with CPO without linesearch is mainly because linesearch in TRPO (and CPO) is generally a technique (that is agnostic to the choice of RL algorithms) to ensure constraint satisfaction as well as performance improvement. It is not only limited to improving the performance of TRPO/CPO. While linesearch can also be used in the Lyapunov-based policy gradient algorithm, it is not a part of the original algorithm. Therefore, for the sake of fair comparisons we remove linesearch in CPO during evaluation. Comparisons with linesearch will be left as future work.
>
> Unclear statements:
> DDPG v.s. PPO: Here the term covariate shift means that the training data is generated by a policy that is different from the current policy that is being trained, i.e., the RL training is done in an off-policy fashion.
>
> Sec 5, second paragraph:
> The robot’s actions are two-dimensional vectors. First dimension is the robot's desired linear velocity (speed at which the robot should go straight). The second dimension is the robot's angular velocity - speed at which the robot should turn. Both velocity vectors are applied on the center of the mass of the robot. This is commonly known in robot kinematics literature as twist (https://en.wikipedia.org/wiki/Robot_kinematics). We will clarify that in the final paper.
>
> Minor points:
> Thank you for catching that. We will correct the typos in the final paper.

---

> > ### Comment · AnonReviewer1 · 2019-11-13
> > **Thank you for the respone**
> >
> >
> > Assuming that you will make the changes promised.  I still have the following concerns
> >
> > I am still sceptical about your comments on CPO comparison. Your comment is circular argument which I can't follow clearly.
> > I am still not sure how constraint satisfaction works without line-search ? Without this a fixed step size may be picked which will always underperform/ violate constraints. In contrast you could pick very small fixed step size ensuring constraint satisfaction but algorithm would be suboptimal.  Neither is  good comparison metric.
> > I also disagree that comparison with line-search is future work, as constraint satisfaction is guaranteed for a given fixed step size.
> >
> > Cheetah Constraints:
> > If empirically results are same I suggest  to use joint constraints. Using velocity constraints can be achieved by massive jerks in place, which by no means is safe planning. In contrast if you can run faster whilst keeping joint movement smooth, this is better performance.

---

> > > ### Author Response · Authors · 2019-11-14
> > > **Additional response to your concerns**
> > >
> > > Thank you for your additional comments/feedback. We have incorporated the changes you suggested in your initial review, except those regarding rewriting Sections 2 and 3.1, in the new version of the paper (and we will fix the page limit once everything is updated). We will improve the readability of these two sections and make them more self-contained in the final version of the paper. Below is our response to your two new comments.
> > >
> > >
> > > CPO and Linesearch
> > >
> > > We agree with the reviewer that (backtracking) linesearch can help with enforcing the constraints and improving the balance between constraint satisfaction and performance. Yet, unless we have access to the true gradient in SGD-based policy gradient (PG) methods, such as TRPO, CPO, and our Lyapunov-based algorithms, linesearch still cannot guarantee constraint satisfaction. This is why CPO, a TRPO-based algorithm, in addition to linesearch, uses other techniques, such as safe-guard policy update and cost shaping (see Section 6.2 and 6.3 of the CPO paper https://arxiv.org/abs/1705.10528), to enforce constraint satisfaction. It is important to note that both linesearch and these additional techniques are agnostic to CPO and can be used together with our Lyapunov PG algorithms. Therefore, for the sake of clarity and fairness in comparing different algorithms, we decided to remove all the techniques used to improve the performance/constraint satisfaction balance from the algorithms, and conduct our experiments with their vanilla version. In order for all methods to be consistent in terms performance-feasibility tradeoff, for fixed learning rates, we run grid-search with the same intervals to choose the best learning rate for each method.
> > >
> > > We would like to emphasize that the vanilla version of CPO is equivalent to our Lyapunov-based PG algorithm with \theta-projection (as shown in the paper). We would leave further improving our Lyapunov-based algorithms by adding techniques such as line search, safe-guard policy update, cost shaping, etc., and experimenting with them in larger scale problems for future work.
> > >
> > >
> > > HalfCheetah constraint:
> > >
> > > We agree with the reviewer that joint torque limit is an intuitive constraint. In fact, both joint torque limit and velocity constraints are well-motivated and have been shown to be useful in different applications (please find more details in Section 3 of https://arxiv.org/pdf/1904.12901.pdf). While the joint torque limit is an action-based constraint, which can alternatively be enforced by specific policy parameterizations (e.g., see Appendix C of the SAC paper https://arxiv.org/abs/1801.01290), the velocity constraint is a kinematic state-based constraint that depends on the state transitions of the MDP. Aside from tackling specific robot control problems, the objective of the benchmark MuJoCo (including HalfCheetah) experiments is to demonstrate/compare the effectiveness of different safe RL algorithms in solving CMDPs. Therefore, we decide to test our algorithms with the latter choice because it cannot be easily formulated without the CMDP framework.

---

> > > > ### Comment · AnonReviewer1 · 2019-11-15
> > > > **Thanks for the response**
> > > >
> > > > Yes, editing section 3.1 is in your best interest as it was the hardest section for me to follow in an otherwise well argued paper.
> > > >
> > > > Line Search:
> > > >
> > > > Okay, that sounds fair. Thank you for the clarification. Maybe make this clear somewhere? Appendix is also fine as these are finer details and you are already running out of page limit and I feel this subtlety useful information for the community as well.
> > > >
> > > > HalfCheetah:
> > > >
> > > > I think there is some miscommunication, from your explanation it sounds like you are talking about joint velocities and not the velocity of the Cheetah as a whole. I might have misread this then.
> > > > Yes, joint velocity as the reference you cited proposes is perfectly reasonable constraint.  From  your description in the paper it sounds more like the restriction is on the velocity of the cheetah as a whole, which I find  counterintuitive to the task (run as fast as possible). If it joint velocities, can you add this ? In which case it is perfectly reasonable to expect torque and velocity constraint to achieve similar performance in the final result.
> > > > If it is not joint velocities , I am afraid we have to agree to disagree on this one and I wouldn't hold this against you.

---

### Official Review · AnonReviewer3 · 2019-10-23
**Official Blind Review #3**

**Rating:** 8

**Review:**

The paper presents a technique for extending existing reinforcement learning algorithms to ensure safety of generated trajectories in terms of not violating given constraints.

I have very little knowledge of this area and as a result was not able to evaluate the paper thoroughly. However, the problem addressed is certainly a very important one and based on my high-level understanding of the concepts involved the approach seems sensible. The experiments are clear and well designed, showing the trade-off between performance and safety.

**Experience Assessment:**

I do not know much about this area.

**Review Assessment: Checking Correctness Of Derivations And Theory:**

I did not assess the derivations or theory.

**Review Assessment: Checking Correctness Of Experiments:**

I assessed the sensibility of the experiments.

**Review Assessment: Thoroughness In Paper Reading:**

I made a quick assessment of this paper.

---

> ### Author Response · Authors · 2019-11-07
> **Thank you**
>
> We thank the reviewer for appreciating our work in terms of novelty, theory, and experiments.

---

### Official Review · AnonReviewer2 · 2019-10-28
**Official Blind Review #2**

**Rating:** 6

**Review:**

This is a very complete submission. There is a novel analysis,
simulations, as well as some results on real data. The authors propose
Lyapunov-based safe RL algorithms that can handle
problems with large or infinite action spaces, and return safe
policies both during training and at
convergence. As far as I can tell the approach is novel, makes sense,
and requires a lot of technical innovations. I was impressed with the
method and the analysis behind the method. The incorporation of the
Lyapunov idea from control theory makes a great deal of sense in this
application. However, it is not trivial to get from using this tool to
a working method.

**Experience Assessment:**

I have published one or two papers in this area.

**Review Assessment: Checking Correctness Of Derivations And Theory:**

I assessed the sensibility of the derivations and theory.

**Review Assessment: Checking Correctness Of Experiments:**

I assessed the sensibility of the experiments.

**Review Assessment: Thoroughness In Paper Reading:**

I read the paper thoroughly.

---

> ### Author Response · Authors · 2019-11-07
> **Thank you.**
>
> We thank the reviewer for appreciating our work of deriving a novel, Lyapunov-based approach to enforce safety in reinforcement learning (RL) algorithms, and the effort of making these algorithms work in practice (in both the MuJoCo benchmark experiments and the indoor robot navigation example).

---

### Author Response · Authors · 2019-12-23
**Response to the meta-reviews**

Since the issues raised in the meta-review are new and not brought up by the three reviewers, we address them here, as we would if they were brought up during the discussion period.

“KL divergence between parameters”: This is a shorthand notation on the KL divergence between policies, which is defined in the end of section 2.1

“Constraint violation for some trajectories”, "achieve safe learning": The notion of safety studied in this paper is based on the EXPECTED cumulative constraint cost that lies below a threshold. This is defined in Section 2, which was motivated detailedly in the second and third paragraphs of the introduction section when CMDPs are introduced. We never claimed that our methods guarantee constraint satisfaction for any trajectory. Our experiments corroborate the results of our proposed algorithm on achieving safe learning (under our defined notion of safety).

“More honest discussions”:  In Section 3, we explicitly describe how the Lyapunov-based PG is derived, including how function approximation and linear approximation are involved. In Section 4, Section 5, Appendix D and E, we provide details on the experimental results on both the MuJoCo and the robot navigation tasks, including comparisons with CPO, the SOTA method (also see response to Reviewer 1 for more details), clear descriptions on how additional safeguard policy update techniques are required in the presence of function approximation (see Appendix C.4 and Appendix D.1 for details). In appendix A to C, we provide discussions and derivations on the baseline Lagrangian algorithm, the Lyapunov approach, and the propositions in the paper. Instead of giving a subjective comment, it would be helpful if the meta-reviewer elaborated on which angle would he/she think our paper is lacking “honest discussions”?

“Safety guarantees”: That is a valid comment. However providing explicit safety guarantees is difficult in most model-free methods with function approximations. While the original Lyapunov-approach (from Chow et al., NIPS 2018) for MDP planning does have explicit safety guarantees, such guarantees are no longer valid for both value-based RL and policy gradient. That being said, similar issues do exist in other model-free RL baselines such as Lagrangian method and CPO, because with function approximation, keeping track of whether the constraint is satisfied requires studying the “size” of this function space (either via VC dimension or Radamacher complexity). In practice, even if we analyze such a bound, it would be of theoretical interest and by most practical means it’s overly-conservative to implement the exact algorithm (similar to TRPO). On the other hand, in this work we empirically showed that the Lyapunov-based algorithms achieved good performance in terms of balancing learning and safety guarantees in both synthetic and real-world examples, and we believe such a contribution has important implications to RL applications. We did include similar discussions (see Appendix C.4 and D.1) in the paper and defer the mathematical derivations of safety guarantees as future work.

“Allow for any collisions in robot experiments”: We clearly described the problem formulation of our robot experiment in the second paragraph of Section 5, in which constraint is based on collision energy (which depends on collision speed). To reduce the conservativeness of the safe RL algorithms, we allow the robot to brush off objects (such as wall) but do not allow it to ramp into obstacles that damage the system for safety reasons.

“Robotics experiment, Threshold in the paper looks pretty arbitrary”: This threshold is based on the maximum allowable collision energy of the Fetch robot.

"Better data efficiency": Compared with existing SOTA such as CPO, our Lyapunov-based safe RL approach is more general and can be applied in off-policy RL algorithms. Discussions can readily be found in the last paragraph of Section 4.

"Scalable to tackle real-world robotics problems": Our algorithm is trained on the robot simulator (see Figure 5 for details) and is only deployed in the real-world environment for evaluation, which does not require a large amount of real-world trials.

---

### Author Response · Authors · 2019-12-23
**Issues with the Reviewing Process of ICLR Paper: Safe Policy Learning for Continuous Control**

Dear ICLR-2020 Program Chairs,

Thank you for all your efforts in organizing a conference at the scale of ICLR.

We are writing to you in regards to our submission entitled: "Safe Policy Learning for Continuous Control", which was rejected despite receiving an average score of 6.67. We present a rebuttal to the meta-review, and kindly ask for reconsideration.

The paper received scores of 8,6,6. The two reviewers, who gave the paper score of 6, R1 and R2, had high confidence (having published one or two papers in this area), and the one with a score of 8, R3, had low confidence. During the rebuttal phase (ended on November 15th, 2019), we responded to each reviewer’s questions in detail and addressed most of their concerns. In particular, after our responses and an exchange with one of the knowledgeable reviewers (R1), she/he seemed to be pleased with our response and maintained her/his vote for acceptance.

However, the meta-review, released on December 19th, 2019, effectively ignored all three reviews/scores, offered another independent review, and decided to reject the paper without giving us an opportunity to address her/his concerns. This poses two issues.

First, the resulting review process for this paper was inconsistent with the ICLR review guidelines and philosophy, and that placed our paper in a disadvantaged position. The AC begins the meta-review by saying “I ignore R3 because that review is useless”. All the reviews were posted on Nov. 5th, over a month and a half ago. It is the AC’s responsibility to read the reviews and if she/he finds a review “useless” (in her/his words) to invite another reviewer or write a review her/himself. To provide the authors with enough feedback and time to respond to the issues raised in this new review, and to treat all the papers equitably, the additional reviews should be made available before the end of the rebuttal phase. None of this was done by this particular AC. This is especially important in a conference like ICLR that uses OpenReview and essentially advocates for more discussions between authors and reviewers (then other conferences with the standard review process).

Second, it is very important to note that the AC’s rejection recommendation was not based on a sudden discovery of a fundamental flaw in the paper, missed by the reviewers, such as error in proof for a theoretical paper that would nullify all the conclusions and contributions, or a discovery of similar work. Such a case would warrant a sudden rejection. Instead, this is a practical, algorithmic paper, and all the issues raised by the AC are subjective in nature, based on her/his reading the paper and the evidence and experiments we provided to support our algorithms. Moreover, three reviewers, two of them experts in the area, did not have the same read of our results as the AC.

In fact, out of 303 papers with scores of 6.67 or higher, only ten were rejected. The rejections were due to close similarity with other work (1, 2), unresolved reviewers’ concerns (3, 4, 5, 6, 7, 8). And, in one case (9), AC posted a review before November 15, giving authors an opportunity to respond, which they did. Our paper is the only rejected paper with a score of 6.67+ purely on the basis of a meta review without a chance to respond. Since the issues raised in the meta-review are new and not brought up by the three reviewers, we address them separately in the tab: "Response to the meta-reviews" , as we would if they were brought up during the discussion period.

We, the authors of the ICLR submission entitled “Safe Policy Learning for Continuous Control”, are all active members of the ML community, have published papers and served as reviewers and ACs at top ML venues for several years. We all share, enforce, and applaud a rigorous and fair review process, and know firsthand how difficult these decisions can be, especially when the field is growing this rapidly. From that perspective, the review of our paper seems like an unusual situation that warrants a closer look.

Given the circumstances of receiving a meta-review that a) overrules all reviewers’ unanimous recommendations of acceptance (and/or weak acceptance), b) raises new concerns without giving us the opportunity to respond, and c) the new concerns are subjective in nature and not due to a discovery a foundational flaw in the paper, we kindly ask the program chairs and/or the senior area chair to take into consideration our response to the meta-review and reevaluate the decision. Thank you all for your consideration.


Best Regards,
Authors of ICLR Paper: Safe Policy Learning for Continuous Control

---

### Decision · Program_Chairs · 2019-12-19

**Decision:**

Reject

**Comment:**

The paper is about learning policies in RL while ensuring safety (avoid constraint violations) during training and testing.

For this meta review, I ignore Reviewer #3 because that review is useless. The discussion between the authors and Reviewer #1 was useful.

Overall, the paper introduces an interesting idea, and the wider context (safe learning) is very relevant. However, I also have some concerns.
One of my biggest concerns is that the method proposed here relies heavily on linearizations to deal with nonlinearities. However, the fact that this leads to approximation errors is not being acknowledged much. There are also small things, such as the (average) KL divergence between parameters, which makes no sense to me because the parameters don't have distributions (section 3.1).

In terms of experiments, I appreciate that the authors tested the proposed method on multiple environments. The results, however, show that safety cannot be guaranteed. For example, in Figure 1(c), SDDPG clearly violates the constraints. The figures are also misleading because they show the summary statistics of the trajectories (mean and standard deviation). If we were to look at individual trajectories, we would find trajectories that violate the constraints. This fact is brushed under the carpet in the evaluation, and the paper even claims that "our algorithms quickly stabilize the constraint cost below the threshold". This may be true on average, but not for all trajectories. A more careful analysis and a more honest discussion would have been useful. In the robotics experiment, I would like to understand why we allow for any collisions. Why can't we set $d_0=0$, thereby disallowing for collisions. The threshold in the paper looks pretty arbitrary.  Again, the paper states that  "Figure 4a and Figure 4b show that the Lyapunov-based PG algorithms have higher success rates". This is a pretty optimistic interpretation of the figure given the size of the error bars.

There are some points in the conclusion, I also disagree with:
1) "achieve safe learning": Given that some trajectories violate the constraints, "safe" is maybe a bit of an overstatement
2) "better data efficiency": compared to what?
3) "scalable to tackle real-world problems": I disagree with this one as well because for all experiments you will need to run an excessive number of trials, which will not be feasible on a real-world system (assuming we are talking about robots).

Overall, I think the paper has some potential, but it needs some more careful theoretical analysis (e.g., effect of linearization errors) and some better empirical analysis.

Additionally, given that the paper is at around 9 pages (including the figures in the appendix, which the main paper cites), we are supposed to have higher standards on acceptance than an 8-pages paper.

Therefore, I recommend to reject this paper.